# A bioinspired scaffold for rapid oxygenation of cell encapsulation systems

Long-Hai Wang[1], Alexander Ulrich Ernst [1], Duo An [1], Ashim Kumar Datta[1], Boris Epel[2], Mrignayani Kotecha[3] & Minglin Ma [1✉]

Inadequate oxygenation is a major challenge in cell encapsulation, a therapy which holds potential to treat many diseases including type I diabetes. In such systems, cellular oxygen ($O_2$) delivery is limited to slow passive diffusion from transplantation sites through the poorly $O_2$-soluble encapsulating matrix, usually a hydrogel. This constrains the maximum permitted distance between the encapsulated cells and host site to within a few hundred micrometers to ensure cellular function. Inspired by the natural gas-phase tracheal $O_2$ delivery system of insects, we present herein the design of a biomimetic scaffold featuring internal continuous air channels endowed with 10,000-fold higher $O_2$ diffusivity than hydrogels. We incorporate the scaffold into a bulk hydrogel containing cells, which facilitates rapid $O_2$ transport through the whole system to cells several millimeters away from the device-host boundary. A computational model, validated by in vitro analysis, predicts that cells and islets maintain high viability even in a thick (6.6 mm) device. Finally, the therapeutic potential of the device is demonstrated through the correction of diabetes in immunocompetent mice using rat islets for over 6 months.

[1] Biological and Environmental Engineering, Cornell University, Ithaca, NY, USA. [2] Department of Radiation and Cellular Oncology, The University of Chicago, Chicago, IL, USA. [3] O2M Technologies, LLC, Chicago, IL, USA. ✉email: mm826@cornell.edu

Cell-based therapies are attractive treatments for a variety of diseases, such as diabetes[1,2], liver diseases[3], and hemophilia[4,5]. In particular, the delivery of islets (or stem cell-derived β-cells) represents a promising therapy for type 1 diabetes (T1D). Islet transplantation via the injection of isolated islets into the liver portal vein or onto the omentum has shown the potential to normalize glycemic control without exogenous insulin in clinical trials, but life-long recipient immunosuppression is required with this procedure[6,7].

Cell encapsulation technology offers to protect cells from immune rejection by isolating them from the host using an artificial, semipermeable material, thereby overcoming the need for immunosuppressive agents[2,8]. Unlike traditional organ transplantations (e.g., pancreas transplantation), wherein the host circulatory system is connected to the transplanted organ via surgical vascular anastomosis[9], most islet encapsulation devices (i.e., bioartificial pancreases) remain isolated from the host's bloodstream after transplantation. Therefore, encapsulated cells are entirely dependent on $O_2$ and other nutrients by passive diffusion from the surrounding blood vessels at the exterior of the device[10]. It is well documented that $O_2$ is more severely limited than other nutrients because of the relative scarcity of extravascular $O_2$ in vivo[11]. In the intraperitoneal site, the $O_2$ tension ($pO_2$) is approximately 40 mmHg, and in the subcutaneous site, likely lower[12].

A thoroughly investigated approach to improve graft oxygenation is to supply exogenous $O_2$ in situ. The βAir device (Beta-O2), for example, supports injections of high concentration $O_2$ into a gas-permeable chamber adjacent to hydrogel-encapsulated cells[13,14]. Another strategy is the local production of $O_2$ using chemical reactions[15,16] or electrolysis[17,18]. Though these strategies have all demonstrated the benefit of adequate $O_2$ supply to encapsulated cells, remaining limitations include increased device complexity and the requirement of patient compliance to maintain $O_2$ provision. $O_2$ transport in hydrogels is invariably dependent on its permeability, the product of the solubility and diffusivity coefficients, both of which are low in aqueous media such as hydrogels and tissue. An alternative, possibly simpler or complementary approach is thus to improve the $O_2$ permeability of the encapsulating material.

In the absence of supplemental $O_2$ provision, theoretical analyses suggest that islets should be within a few hundred micrometers from the bloodstream in surrounding tissue to avoid hypoxia[19,20]. Based on this design principle, the cell module of an encapsulation system should be exceedingly thin to support favorable oxygenation[12,21]. For spherical microcapsules which are endowed with a high surface area to volume ratio, a diameter of ~1000 μm is widely used[11,22,23]. Similarly, cylindrical cell-laden hydrogel fibers are commonly designed from 350–1000 μm in diameter[24–26], and planar slabs (the geometry endowed with the lowest surface area to volume ratio), typically from 250–600 μm in thickness[8,13,14].

Clinical islet transplantations require approximately 500 k islet equivalent (IEQ) of human islets (5–10 k IEQ per kg body weight) to reverse diabetes[1], and cellular treatments for liver diseases and hemophilia require similar cell volumes[5]. Because $O_2$ diffusion limitations restrict hydrogel thickness, devices can only be scaled along one or two spatial dimensions to accommodate this requisite cell payload, and thus an unreasonably large estimated device length, surface area, or number is required. It is estimated that meters of a cylindrical fiber, hundreds of square centimeters of a planar slab, or ~100,000 microcapsules are needed to deliver a curative islet dose. Increasing the system's $O_2$ permeability would allow devices to be scaled in three dimensions, facilitating the design of reasonable and surgically convenient device geometries (i.e., shorter in length, smaller in surface area, or lower in number).

The physiology of insects presents a creative solution to rapid $O_2$ distribution across multi-millimeter scales. Instead of using circulatory blood for tissue oxygenation as in vertebrates[27], insects transport gaseous oxygen via a gas-filled channel network, known as the tracheal system, which is distributed throughout their bodies; this system is also present in some aquatic insects without spiracles[28,29]. Inspired by this clever mechanism of rapid gas-phase $O_2$ distribution, we pursued the design of an air-filled scaffold for islet encapsulation to overcome the thickness limitation for $O_2$ diffusion, which we name SONIC (Speedy Oxygenation Network for Islet Constructs) (Fig. 1).

We highlight the bio-inspiration by analyzing the tracheal anatomy of a larva of the mealworm beetle (*Tenebrio molitor*) (Fig. 1a), which guided the design of the SONIC scaffold. High-resolution X-ray computer tomography (Nano-CT) scanning was used to visualize the tracheal network in the mealworm (Supplementary Fig. 1), showing its thorough distribution throughout the body in a ladder-like geometry[30,31] (Fig. 1b). It is evident from this pattern that $O_2$ is delivered to insect tissue through the air channels in the tracheae (Fig. 1c). Importantly, the $O_2$ diffusion coefficient in the gaseous phase is around $10^4$ times higher than that in water or biological tissue[29,32]. Thus, the tracheal network leverages rapid gas-phase $O_2$ transport, allowing insects to efficiently distribute $O_2$ throughout their bodies[33]. This innovative $O_2$ transport mechanism allows some insects to grow to incredibly large sizes[34], such as the Goliath beetle larvae reaching the size of a mouse, the *Dynastes Hercules* larvae reaching 147 mm long with a weight of 144 g, and the *Megasoma* larvae reaching a maximum dorsal width of 225 mm[35].

Herein, the incorporation of the SONIC scaffold in a hydrogel-based cell encapsulation system intentionally recapitulates the efficiency of rapid $O_2$ transport of the insect tracheal network, oxygenating deeply encapsulated cells within thick devices (Fig. 1d). We confirm the rapid and penetrating $O_2$ transport capability through the SONIC scaffold visually using an $O_2$ imager. Additionally, we demonstrate that the SONIC device can maintain high cell viability and robust function of rat islets in immunocompetent diabetic mice for over 6 months. The biomimetic SONIC cell delivery system solves the problem of slow and non-penetrating $O_2$ transport in thick bulk hydrogels.

## Results

**Design and fabrication of the SONIC device**. To develop a SONIC scaffold suitable for cell encapsulation and supporting of rapid $O_2$ transport, two features, difficult to achieve simultaneously, were essential: (1) an internal microstructure that was bicontinuous and non-wettable to provide unobstructed microchannels for $O_2$ flow similar to the insect tracheae and (2) an external surface that was wettable to allow cell-laden hydrogel precursor to infiltrate. We conjectured that a hydrophobic material for the scaffold skeleton would resist water infiltration. We designed a two-step process, forming first a superhydrophobic (i.e., low surface energy and high roughness) scaffold skeleton with continuous internal pores and then an adherent hydrophilic coating only on the external surface.

Accordingly, poly(vinylidene fluoride-*co*-hexafluoropropylene) (PVDF-HFP) (Fig. 1e) was selected as the skeleton material, which imparts hydrophobicity via the highly fluorinated -$CF_3$ group of the HFP unit[36], while providing high solubility in polar solvents via the unique polarity of the alternating -$CH_2$- and -$CF_2$- groups of the VDF unit along the polymer chain[37]. PVDF-HFP solution could then be subjected to an immersion

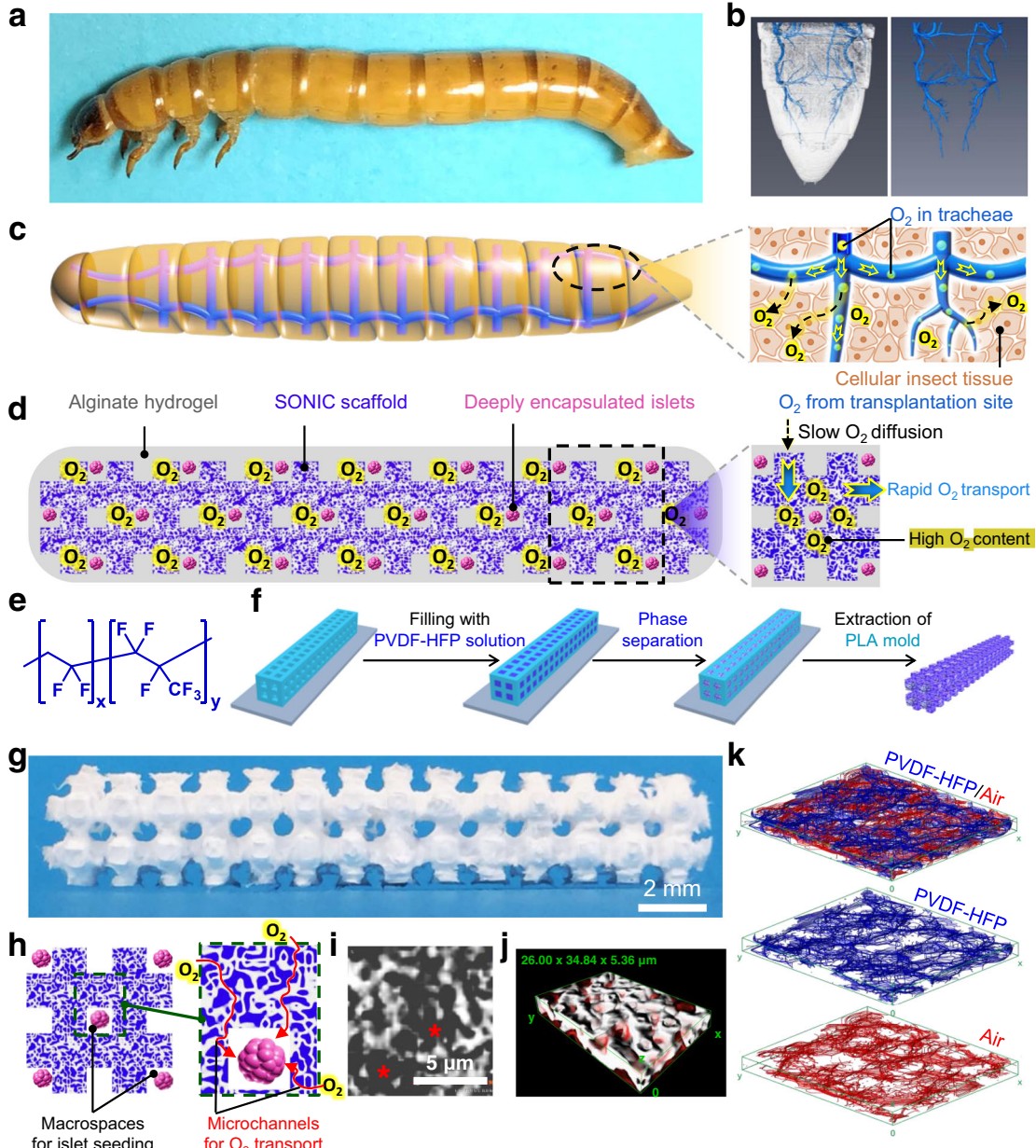

**Fig. 1 Design and fabrication of the biomimetic SONIC scaffold. a** A digital image of a larva of the mealworm beetle (*Tenebrio molitor*). **b** 3D reconstruction of Nano-CT images of the tail of a mealworm (left) and the segmented gas-phase tracheal system (right) inside the body. **c** Schematic illustrating the tracheal system in a mealworm and $O_2$ delivery to the surrounding cellular tissue through the tracheae. **d** A schematic illustrating $O_2$ delivery from the transplantation site into the cell encapsulation system through a tracheal ladder network-like SONIC scaffold. **e** Chemical structure of the fluoropolymer PVDF-HFP. **f** Fabrication of the ladder-like SONIC scaffold. **g** A digital image of the SONIC scaffold. **h** Schematics representing the macro- and microarchitecture structure of the SONIC scaffold. **i** A Nano-CT image of the porous scaffold (the asterisks indicate the pore regions). **j** 3D reconstruction of Nano-CT images of a selected region (26 × 34.84 × 5.36 μm) inside the SONIC scaffold showing the bicontinuous microstructure (the diffuse red coloring indicates the air phase in the porous skeleton). **k** Skeletal networks for the polymeric (blue) and the porous (red) regions of the SONIC scaffold.

precipitation process to produce the interconnected porous and rough microarchitecture via phase separation[38,39]. Furthermore, PVDF-HFP is resistant to hydrolytic, oxidative, and enzymatic breakdown[40], and is therefore advantageous for use in vivo.

The SONIC scaffold was fabricated as shown in Fig. 1f. A multilayer poly(lactic acid) (PLA) mold (Supplementary Fig. 2), complementary to the targeted ladder network geometry (Fig. 1d), was first 3D printed. Subsequently, a PVDF-HFP solution (15 wt % in acetone) was filled into the PLA mold and immersed in a water/ethanol (v/v = 1/1) bath for the phase separation process

yielding the internal porous microstructure containing polymer-rich regions and polymer-poor regions (i.e., the air channels). Finally, the ladder-like PVDF-HFP scaffold was isolated after selective extraction of the PLA mold using chloroform (Fig. 1g and Supplementary Movie 1).

The structural analysis confirmed that the desired macro- and microarchitecture (Fig. 1h) of the SONIC scaffold was achieved. Nano-CT imaging illustrated an internal microporous structure with a pore size of a few micrometers (Fig. 1i). Additionally, renderings of 3D reconstruction images revealed that the internal air channels were

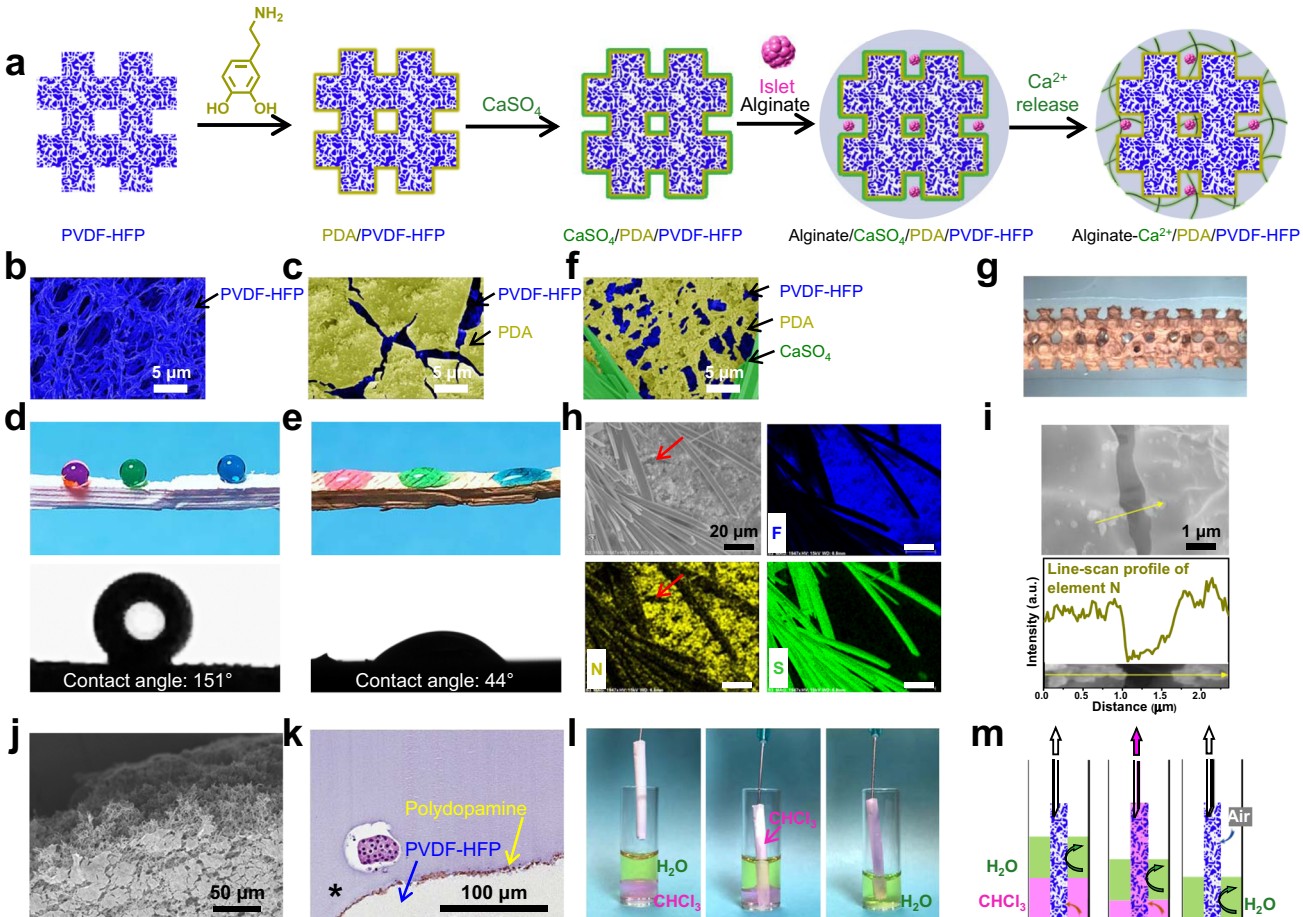

**Fig. 2 Fabrication and characterizations of the SONIC device. a** Schematic illustration of the device fabrication (side view). **b, c** False-colored SEM images of the SONIC scaffold (**b**) and polydopamine-coated SONIC scaffold (**c**). One representative of 3 independent experiments is shown. **d, e** Digital images of water droplets (colored with food dye) and contact angle goniometer-captured images of a water droplet on a rectangular prism SONIC scaffold before (**d**) and after (**e**) polydopamine modification. **f** A false-colored SEM image of the polydopamine-coated SONIC scaffold with deposited $CaSO_4$ crystals. One representative of 3 independent experiments is shown. **g** Stereo microscope image of the SONIC device (top view). **h** SEM/EDS elemental mapping of F, N, and S on a polydopamine-coated SONIC scaffold with deposited $CaSO_4$ crystals. The red arrows indicate the lack of polydopamine at a coating crack location. One representative of 3 independent experiments is shown. **i** SEM image of the polydopamine-coated SONIC scaffold and the corresponding element N distribution profile across a polydopamine coating crack. One representative of 3 independent experiments is shown. **j** SEM image of the cross-sectional polydopamine-modified SONIC scaffold, showing no polydopamine inside the scaffold. One representative of 3 independent experiments is shown. **k** An H&E staining slice of an islet encapsulation device showing the polydopamine located at the interface between the SONIC scaffold and alginate hydrogel. One representative of 10 replicates is shown. **l** Captured images during the perfusion test using a cylindrical SONIC scaffold. A pump-connected needle was inserted into one end of a cylindrical SONIC scaffold, and the other end of the SONIC scaffold was immersed into a vial containing water phase (top, colored with green food dye) and chloroform phase (bottom, colored by Nile Red dye). **m** Schematics representing the distribution of water and chloroform during the perfusion test.

indeed continuous (Fig. 1j, Supplementary Fig. 3, and Supplementary Movie 2). Furthermore, skeletal network analysis for the polymeric skeleton and the air channels confirmed the bicontinuous structure (Fig. 1k). These characterizations confirmed that the SONIC scaffold recapitulated the essential features of the tracheal system necessary for rapid gas-phase $O_2$ transport.

To integrate the SONIC scaffold into a hydrogel-based cell encapsulation device (Fig. 2a), it was critical to render the scaffold surface wettable. Dopamine can self-polymerize to form a surface-adherent hydrophilic polydopamine film onto a wide range of substrates, including on hydrophobic surfaces[41]. Thus, we applied the mussel-inspired hydrophilic polydopamine coating on the scaffold's surface (Fig. 2b, c and Supplementary Fig. 4) but not the internal pores. To test the efficacy of this process, water contact angle (CA) measurements were performed to characterize the wettability of the SONIC scaffold before and after polydopamine modification. The CA of polydopamine-coated

SONIC scaffold decreased from the original 151° (Fig. 2d and Supplementary Movie 3) to 44° (Fig. 2e), showing the substantially increased wettability.

Following surface hydrophilicity modification, a cell-laden hydrogel was applied via a simple in situ procedure. $CaSO_4$ was deposited onto the SONIC scaffold surface as a crosslinker source (Fig. 2f) for the following application of a cell-containing alginate solution (SLG 100, 2 wt%). In comparison with calcium salt ($CaCl_2$), which is more commonly used for alginate cross-linking, $CaSO_4$ has a much lower solubility[42], retarding the gelation rate to provide time for the alginate solution to penetrate into the scaffold skeleton before gelling (Fig. 2g). Furthermore, the slow dissociation of $Ca^{2+}$ from the $CaSO_4$ crystals produces a more uniform and mechanically stronger alginate hydrogel than fast gelation in a $CaCl_2$ bath[43].

Scanning electron microscopy (SEM)/energy-dispersive X-ray spectroscopy (EDS) mapping was used to analyze the component

distribution on the SONIC scaffold. Fluorine (F), nitrogen (N), and sulfur (S) were chosen as the specific elements to identify PVDF-HFP, polydopamine (PDA), and CaSO₄, respectively (Fig. 2h). The SEM/EDS line profile of the N signal across a coating crack showed the lack of polydopamine inside the scaffold (Fig. 2i). In addition, an SEM cross-sectional image of the polydopamine-modified scaffold showed that the polydopamine coating was only distributed on the surface (Fig. 2j). Furthermore, hematoxylin and eosin (H&E) staining of a slice of an islet encapsulation device also confirmed that the polydopamine was only located at the interface between the scaffold and alginate hydrogel (Fig. 2k). These characterizations confirmed that hydrophilicity modification was limited to the scaffold surface and therefore unlikely to affect the hydrophobicity of internal regions of the scaffold.

We performed a perfusion study to test the ability of the SONIC scaffold to prevent water from wicking into the internal pores. A cylindrical scaffold was inserted into a vial containing a water phase and a chloroform phase (Fig. 2l). The chloroform quickly penetrated the scaffold by capillary action and spread through the whole scaffold by suction force applied on the top, whereas the water was excluded and remained outside the scaffold even after all chloroform was withdrawn (Fig. 2m). The flow of chloroform throughout the scaffold also confirmed the SONIC scaffold's bicontinuous porous structure, validating visual observations in previous Nano-CT scanning characterizations (Fig. 1j, k).

**Rapid O₂ transport through the SONIC scaffold**. O₂ distribution mapping was performed on an electron paramagnetic resonance (EPR)[44,45] O₂ imager to characterize O₂ transport through the SONIC scaffold (Fig. 3). Briefly, a SONIC scaffold or a control PLA filament was inserted in 1% gelatin containing EPR spin probe (OX063-d24, 1 mM)[46] in a glass tube (Fig. 3a–c). First, the system was deoxygenated using N₂ to reduce pO₂ close to 0 mmHg. After deoxygenation, the system was exposed to a gas mixture containing a pO₂ of 40 mmHg, and the pO₂ change in gelatin was continuously monitored until a steady-state (40 mmHg) was approached. The average pO₂ in gelatin with the PLA control insert slowly climbed to ~35 mmHg after 11 h (Fig. 3d). Spatial pO₂ distribution plots showed a gradient descending from top to bottom in the tangential plane and uniform profiles in the transverse plane (Fig. 3e). On the other hand, the average pO₂ in gelatin with the SONIC scaffold rapidly increased to the equilibrium level of 40 mmHg within ~2 h (Fig. 3f). Moreover, spatial pO₂ images in the transverse plane showed a radial gradient emanating from the location of the SONIC scaffold before achieving steady-state, and profiles in the tangential plane revealed high pO₂ levels at the bottom of the sample near the SONIC scaffold, even in the first image collected after exposure to the gas mixture (Fig. 3g).

A computational model, developed to simulate spatial O₂ transfer over time in this system (Supplementary Fig. 5), was in general agreement with experimental results. In silico, the SONIC scaffold distributed O₂ more efficiently through the gelatin in comparison to the PLA control. Likewise, a clear radial O₂ gradient was observed emanating from the SONIC sample, whereas only a top-down gradient was observed in the control PLA simulation (Fig. 3h and Supplementary Movie 4). More importantly, SONIC incorporation was predicted to equilibrate the surrounding gelatin within roughly 2 h, whereas equilibration in the control system was significantly delayed (Fig. 3i and Supplementary Movie Video 4). Mechanistically, this is because diffusive resistance to O₂ transport in the SONIC scaffold is negligible compared to that within the hydrogel, and thus a high pO₂ gradient is established between the SONIC scaffold and the hydrogel at their interface, driving O₂ transport into the hydrogel.

The EPR measurement procedure was repeated with two additional control scaffolds chosen to verify the vital role of the air channels in facilitating high oxygen permeability. One control was a commercial porous sponge comprised of hydrophilic melamine, which was selected as a simple example to confirm that a liquid-filled porous material would not provide benefit to O₂ transport. The second control was a PVDF-HFP scaffold, modified by the following two-step process to render the internal microporous channels hydrophilic: (1) ethanol was added to the tris buffer during incubation with dopamine to allow buffer solution penetration into the micropores of the scaffold, therefore enabling the application the hydrophilic polydopamine coating within the microchannels; (2) the scaffold was treated with radiofrequency plasma to further ensure its hydrophilicity. O₂ transport tests showed that these two scaffolds were significantly slower than SONIC at equilibrating the system to exposed O₂ levels (Supplementary Fig. 6a). Spatial O₂ distributions of the sample containing the hydrophilic porous PVDF-HFP showed a top-to-bottom pO₂ gradient throughout the system until a steady-state was achieved (Supplementary Fig. 6b), rather than the radial gradient emanating from the SONIC insert (Fig. 3g), the latter of which enabled rapid and deep O₂ penetration to the bottom. The results from these two additional controls indicate the necessity of the non-wettability of the internal microchannels for enabling rapid O₂ transport in the SONIC system.

**The SONIC device improves cell survival under hypoxic conditions**. In vitro tests and a complementary theoretical analysis were performed to evaluate the advantages of the SONIC scaffold for improving the viability of hydrogel encapsulated cells (Fig. 4). A computational model was developed to predict the oxygenation of INS-1 cells (2.5 million cells per mL alginate) in the control device and the SONIC device (Supplementary Fig. 7a). The boundary pO₂ was set as 40 mmHg[47] to mimic the pO₂ level in an intraperitoneal cavity (Supplementary Fig. 7b). As expected, the control device showed large hypoxic regions in the center due to the limited O₂ transport in such a thick bulk hydrogel (Fig. 4a), with a steep pO₂ gradient decreasing from 40 mmHg at the device exterior boundary to ~3 mmHg in the device center, corresponding with a drop of O₂ concentration ($c_{O_2}$) from 0.05–0.004 mM (Fig. 4b). On the other hand, all regions in the simulated SONIC device showed sufficient oxygenation throughout the entire device (Fig. 4c). The pO₂ was maintained high over 35 mmHg in the whole SONIC device with only a minor difference between central and surface regions of the cell/hydrogel phase. Correspondingly, the cell/hydrogel phase showed high $c_{O_2}$ between 0.044–0.05 mM, while the SONIC scaffold showed a substantially higher equilibrium $c_{O_2}$ (~1.85 mM) due to the preferential partitioning of O₂ into the gas-containing SONIC scaffold (Fig. 4d). This suggests that the penetrating redistribution of external O₂ by the SONIC scaffold should have a significant positive impact on cell survival, especially for deeply encapsulated cells.

Results from the in vitro test were consistent with model predictions. Cell-containing devices were incubated in a hypoxic incubator with 5% CO₂ and 5% O₂ (i.e., 40 mmHg). After culturing for 48 h, cell viability was evaluated using a live/dead staining kit. Cells in the central region of the control device were dead likely due to hypoxia, with surviving cells limited to a thin region near the surface (Fig. 4e). In contrast, robust cell viability was detected throughout the SONIC device, even in the device center (Fig. 4f). These findings validated model accuracy and more importantly provided proof-of-concept for incorporating

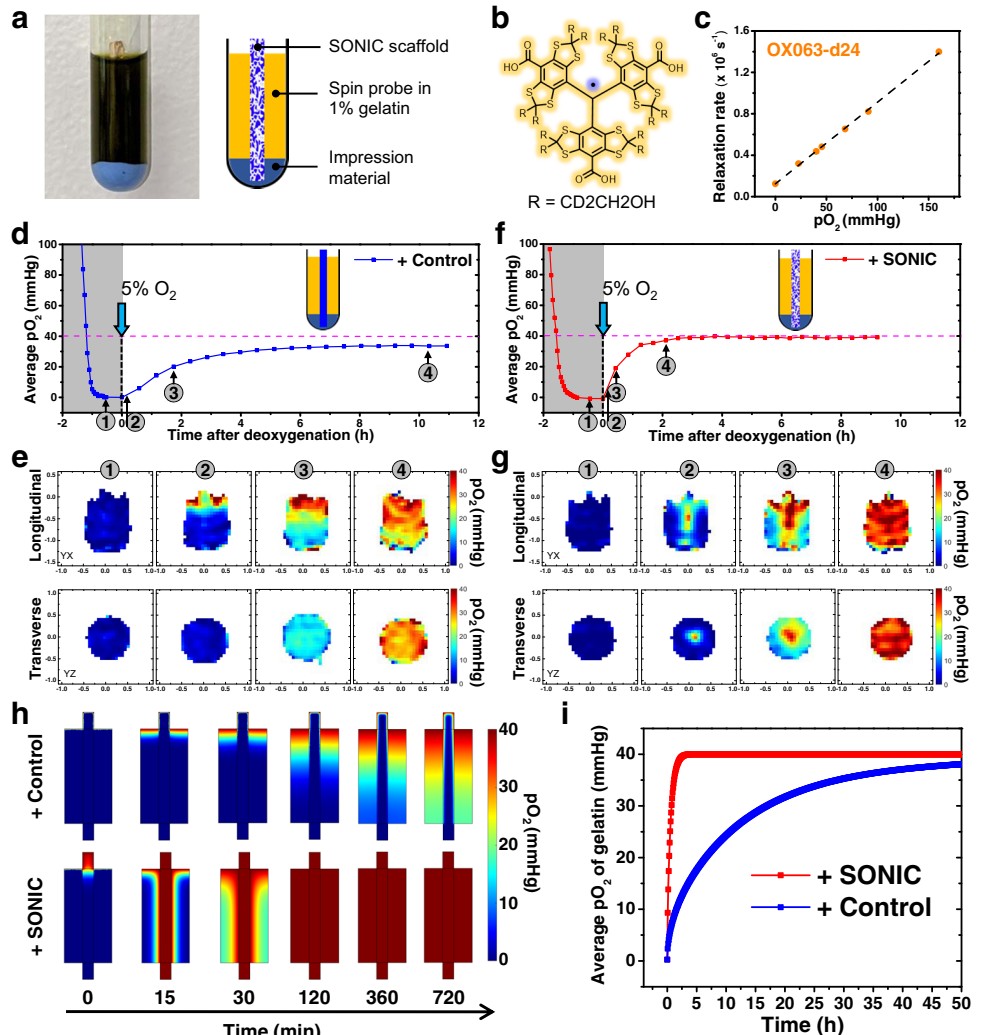

**Fig. 3 Characterizations of rapid O₂ transport through the SONIC scaffold. a** A digital image (left) and schematic (right) showing the sample in a container for the EPR test. **b** Chemical structure of the EPR spin probe OX063-*d24*. **c** Calibration curve of the OX063-*d24* relaxation rate versus pO₂. **d** Average pO₂ of gelatin in the container with a PLA control insert versus time. The initial gray period indicates the deoxygenation of the system. **e** pO₂ distributions on the tangential plane and transverse plane of a sample with the PLA control insert at different time points (indicated by the arrows in Fig. 3d). **f** Average pO₂ of gelatin in the container with a SONIC scaffold versus time showing much faster equilibration. **g** pO₂ distributions on a tangential plane and a transverse plane of the sample with the SONIC scaffold at different time points (indicated by the arrows in Fig. 3f). **h** Simulation data showing spatial pO₂ profiles over time in the system with the PLA control insert and SONIC scaffold. **i** Simulated the average pO₂ of gelatin in the container versus time.

the SONIC scaffold into a device to overcome O₂ diffusion resistance to encapsulated cells.

The computational model was adapted to simulate O₂ profiles in SONIC devices, empty control devices, and control scaffold devices containing islets (Supplementary Fig. 8–10). In this model, 500 IEQ of rat islets were considered as discrete spheres, with diameters (30–300 μm) selected from a distribution—simulating the natural size heterogeneity of islets—and dispersed randomly within the alginate domain (Supplementary Fig. 8a, b). Cross-sectional surface plots show deficient pO₂ levels in the center of the control device (Fig. 4g), whereas pO₂ levels throughout the SONIC scaffold were maintained uniform and high at ~35 mmHg near that of the external source (i.e., 40 mmHg) to provide sufficient O₂ to the surrounding islets (Fig. 4h). Here, SONIC's advantageous O₂ delivery mechanism is illustrated: external O₂ crosses only a thin barrier of the slow-diffusivity alginate before it reaches the SONIC scaffold, where it permeates rapidly throughout the structure, achieving a scaffold

pO₂ level near that of the surrounding environment (Fig. 1d). This rapid equilibration is achieved because of the high O₂ permeability of the SONIC system, which is enabled by the bicontinuous air channels rather than the PVDF-HFP material itself (Supplementary Fig. 8d, e and Supplementary Table 1).

Quantification of the oxygenation of the islet population in devices showed that the average islet pO₂ in the SONIC device was 1.8-fold higher than that in the control device (Fig. 4i), reducing the fraction of necrotic tissue 5-fold compared to islets in the control device (Fig. 4j and Supplementary Fig. 9g, i). Additional analysis indicated that SONIC device encapsulated islets exhibited pO₂ levels descending roughly linear with their increasing diameter (Supplementary Fig. 9c), with only a few of the largest islets displaying small regions falling below pO₂ levels associated with the impaired function (<4 mmHg) or necrosis (<0.08 mmHg)[48,49]. In contrast, many islets, including some small ones, were anoxic in control devices (Supplementary Fig. 9d, e). These results indicate that significantly improving the rate of O₂

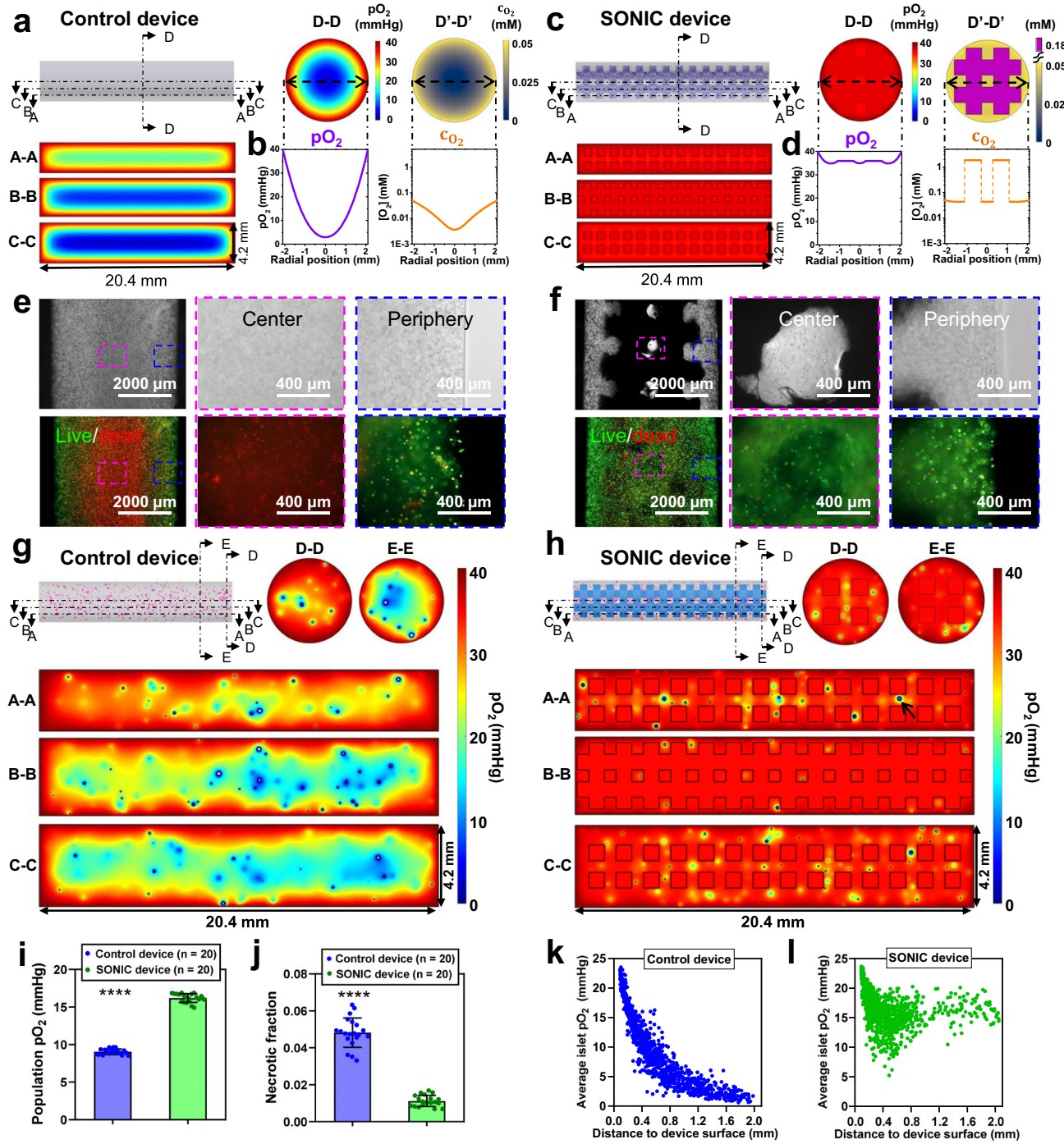

transfer throughout the system bestows substantial benefits to islet oxygenation.

The simulation was repeated, but with the adaptation that each islet was implemented as 150 μm in diameter (the standard size for the definition of an islet equivalent) to isolate the effect of islet distance from the surface (Supplementary Fig. 10). In the control device, it was observed that islets near the device surface were well-oxygenated, while those in the center were anoxic (Fig. 4k). On the other hand, islets at all positions in the SONIC device were reasonably well-oxygenated (Fig. 4l). This shows that islet oxygenation is highly dependent on encapsulation depth in exclusively hydrogel-based systems (e.g., the control device), and that SONIC scaffold incorporation effectively decouples islet

oxygenation from proximity to the external O2 source (i.e., the tissue surrounding the device in the host site).

The in vitro study and theoretical analysis performed here to confirm the benefit of the SONIC scaffold to improve O2 delivery to encapsulated cells and elucidates its mechanism of action. As previously mentioned, it is often suggested that islets must be within a few hundred micrometers to the device-host boundary to avoid debilitating O2 diffusion resistances[19,20]. We predict here that the pO2 levels in the SONIC scaffold are near those available in the transplantation site (i.e., intraperitoneal cavity). The SONIC device was designed such that any islet is within 300 μm from the SONIC scaffold or the device-host boundary. It follows that every encapsulated islet is thus within 300 μm to a

**Fig. 4 The SONIC scaffold improves cell survival under hypoxic conditions. a** Simulation-predicted $pO_2$ distributions of INS-1 cells encapsulated control device in three tangential cross-sections (labeled A-A, B-B, and C-C) and a transverse cross-section (labeled D-D, a corresponding $c_{O_2}$ distribution labeled as D'-D'). **b** Quantitative $pO_2$ and $c_{O_2}$ distributions along a radial line in the transverse cross-section of the control device showing rapid $O_2$ dropping from 40 mmHg (0.05 mM) at the surface to ~3 mmHg (0.004 mM) at the center. **c** Simulation-predicted $pO_2$ distributions of INS-1 cells encapsulated SONIC device in three tangential cross-sections and a transverse cross-section. **d** Quantitative $pO_2$ and $c_{O_2}$ distributions along a radial line in the transverse cross-section of the SONIC device showing high $pO_2$ over 35 mmHg in the whole device, corresponding with high $c_{O_2}$ over 0.044 mM in the cell/hydrogel phase and a substantially higher $c_{O_2}$ of ~1.85 mM in the SONIC scaffold due to the preferential partitioning of $O_2$ into the gas-containing SONIC scaffold. **e, f** Microscope images of live/dead staining of INS-1 cells encapsulated in the control device (**e**) and the SONIC device (**f**). One representative of 2 independent experiments is shown. **g** Simulation-predicted $pO_2$ distributions of rat islets (size-distributed) in a control device in three tangential cross-sections (labeled A–A, B-B, and C-C) and two transverse cross-sections (labeled D-D and E-E) showing massive hypoxic regions in the center of the device with necrosis observed in many islets (white regions in the islets represent necrosis). **h** Simulation-predicted $pO_2$ distributions of rat islets (size-distributed) in a SONIC device in three tangential cross-sections (labeled A-A, B-B, and C-C) and two transverse cross-sections (labeled D–D and E-E) showing well-oxygenated islets in the entire device, with negligible necrosis observed in rare large islets (arrow in A-A cross-section). **i, j** Average $pO_2$ (**i**) and fraction of necrosis (**j**) of the islet populations in control devices ($n = 20$), and SONIC devices ($n = 20$). Mean ± SD, ****$p < 0.0001$ (unpaired two-sided students $t$-test). **k, l** Scatter plots of islet location versus average $pO_2$ in simulated islets (all 150 μm in diameter) in control devices (**k**) and SONIC devices (**l**).

high $pO_2$ source, regardless of its distance to the host site tissue. This not only permits the construction of thick devices, but also should allow the incorporation of significantly higher cell densities without significant impact on cell oxygenation (Supplementary Fig. 7d, e and Supplementary Fig. 11), especially with human islets which feature lower $O_2$ consumption rates[50] (Supplementary Fig. 12), and mitigate negative outcomes should the external $pO_2$ environment be lower (Supplementary Fig. 13). The computational analysis provided here, and corresponding in vitro tests, suggest that the SONIC device significantly improves cell survival in hydrogel-based devices.

**The SONIC device enables long-term diabetes correction in mice.** Following in vitro testing and theoretical analysis, in vivo studies were pursued to test the therapeutic capability of the SONIC device (Fig. 5). Rat islets (500 IEQ per transplant) were incorporated in the SONIC devices (Fig. 5a, b; ~20 mm in length and ~4 mm in diameter) or controls (bulk hydrogels of the same dimensions but without the SONIC scaffold) (Supplementary Fig. 14) and transplanted into the intraperitoneal cavity of diabetic C57BL/6 mice. While both groups achieved normoglycemia within a few days after transplantation, the BG levels in the control group gradually rose and reverted to hyperglycemia after couple of weeks (Fig. 5c). H&E staining of the retrieved control sample showed the islets were necrosed with severe karyorrhexis or complete loss of nuclei (Supplementary Fig. 15). We speculate that the progressive failure of control devices was mainly due to initial hypoxia and necrosis of islets in the center of the device, resulting in the release of danger-associated molecular patterns (DAMPs), and subsequent damage to peripheral islets from DAMP-induced immune activation[51]. In contrast, the BG readings in the SONIC group were under control for over 6 months (Fig. 5c). BG returns to hyperglycemia after device retrieval confirmed that the function of the device was responsible for diabetes correction.

Intraperitoneal glucose tolerance tests (IPGTT) were conducted on day 60 (Fig. 5d) and day 180 (Fig. 5e) before retrievals to evaluate the kinetics of insulin release from the devices. The SONIC group showed a well-preserved glycemic profile similar to that of healthy mice: the BG returned to normoglycemia within 2 h, indicating reasonable glucose clearance by the SONIC devices. By contrast, a poor glycemic profile was observed in the control group, as the BG remained at a high level similar to that of the diabetic control mice. A static glucose-stimulated insulin secretion (GSIS) test of the retrieved SONIC devices also showed glucose responsiveness of the encapsulated islets (Fig. 5f). In addition, the live/dead staining revealed robust islet viability

(Fig. 5g). H&E staining and insulin/glucagon immunostaining further confirmed the viability and function of islets in the retrieved devices. Both the peripheral islets (Fig. 5h) and central islets (Fig. 5i) in the SONIC device showed intact morphology and positive insulin and glucagon expression.

A transverse section of a retrieved device at 6 months confirmed that both the islets close to the surface and deep within the structure were viable (Fig. 5j). This was consistent with a selected transverse surface plot from the computational model, which suggested that islets in both central and peripheral regions would be highly oxygenated (Fig. 5k).

The foreign-body reaction (FBR) is a ubiquitous phenomenon for almost all foreign implants, which induces fibrosis deposition on the implant surface, occasionally leading to a complete coverage by the formation of a collagenous fibrotic capsule[52,53]. Deposited fibrosis restricts the mass transfer at the implant/host interface and affects the viability of the encapsulated cells. Alginate is one of the most widely used materials for cell encapsulation due to its good biocompatibility. In addition, it has been reported that alginate capsules or fibers larger than 1.5 mm in diameter have a significantly mitigated fibrotic response[25,54,55]. The diameter of this SONIC device is around 4 mm which is suggested to mitigate the fibrotic response. In acellular studies, minimal cellular overgrowth was observed on the retrieved device surface (Supplementary Fig. 16). For the islet-containing devices, most regions of the retrieved devices remained free of fibrosis (Fig. 5i, j and Supplementary Fig. 17a), though a mild cellular deposition was observed on partial regions (Fig. 5h, l and Supplementary Fig. 17b–d). Nonetheless, the islets near regions with deposited fibrosis remained healthy and no necrosis was observed (Fig. 5l and Supplementary Fig. 17d). Our computational model was adapted to simulate the effect of fibrosis coverage, implemented by setting a no-flux boundary condition on one half of the device surface, simulating an extreme case where fibrosis is assumed to eliminate $O_2$ transport entirely (Supplementary Fig. 18a). While the simulated fibrotic side of the control device was hypoxic, with islets exhibiting high levels of necrosis, the simulated islets in the SONIC device were adequately oxygenated at both the fibrotic side and the unblocked side (Fig. 5m and Supplementary Fig. 18b). Near homogeneous $pO_2$ levels within the scaffold illustrate that the SONIC scaffold "carries" $O_2$ from high $pO_2$ regions to low $pO_2$ regions over millimetre-scale distances, effectively redistributing the available $O_2$ throughout the device.

**The SONIC system enables diabetes correction in mice with thick device.** Successful in vivo results of the ~4 mm diameter

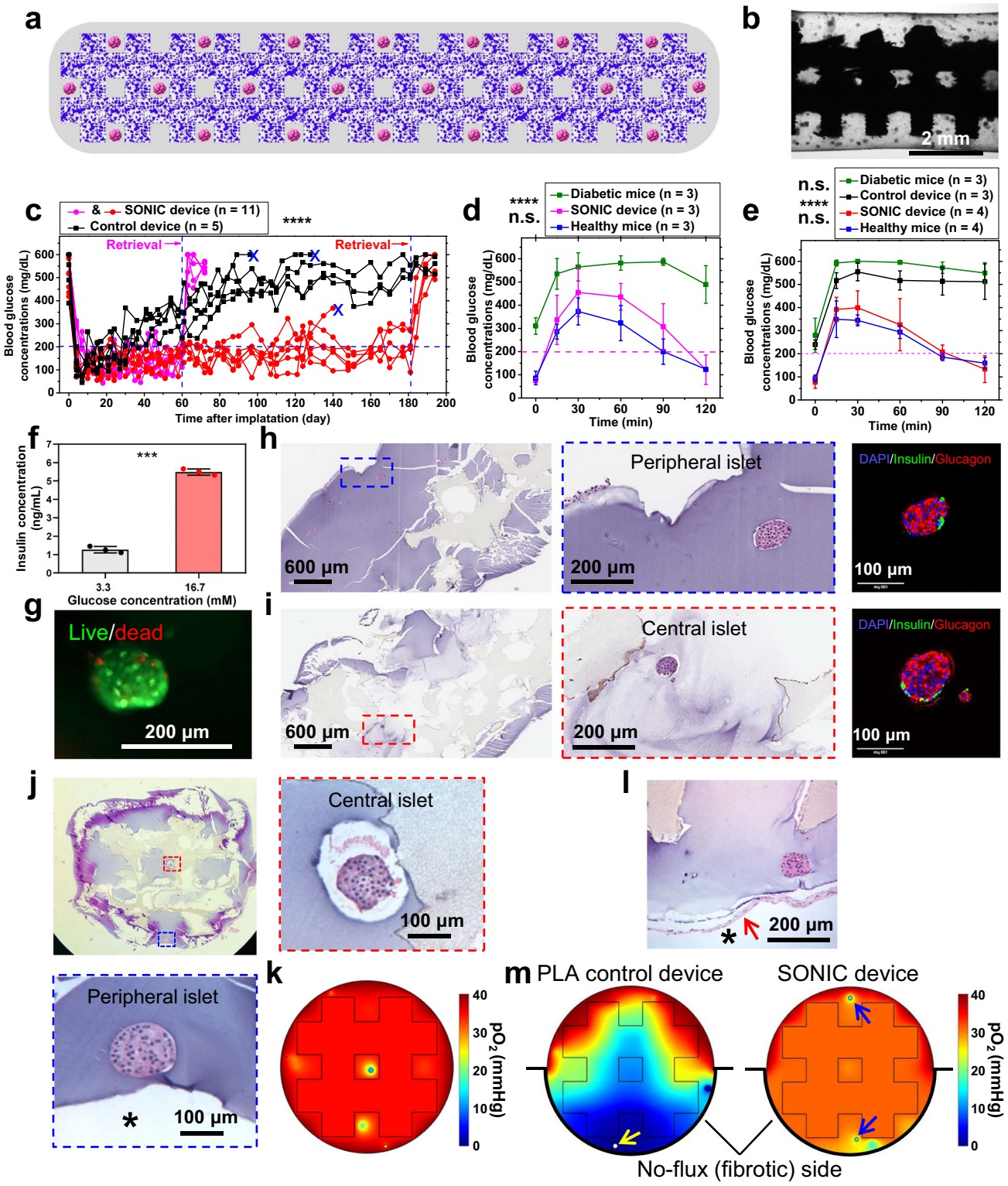

device motivated the exploration of yet a thicker device design (Fig. 6). Our computational model was adapted to simulate the effects of increasing device size prior to fabrication and in vivo testing. A cubic (6.6 × 6.6 × 6.6 mm) control device and SONIC device were considered, each including 500 IEQ of size-distributed rat islets (Fig. 6a, b and Supplementary Fig. 19a). Remarkably, in silico islets in the 6.6 mm-thick SONIC device were similarly oxygenated (with similarly negligible necrosis levels) as the former 4.2 mm device due to the rapid $O_2$

redistribution by the gas-phase transport through the SONIC scaffold (Fig. 6c, d). The simulation, repeated but with all islets being 150 μm in diameter (Supplementary Fig. 19b), showed that the $pO_2$ of control islets decreased with distance from the external boundary (Fig. 6e), whereas even the innermost islets in the SONIC device exhibited high $pO_2$ levels (Fig. 6f). This indicates that the advantage of the SONIC device (decoupling islet oxygenation from distance to the host site) was maintained even at this thicker scale.

**Fig. 5 The SONIC device enables 6-month diabetes correction in mice. a, b** Schematic (**a**) and microscope image (**b**) of rat islets encapsulated in a SONIC device (~4 mm in diameter). One representative of 11 replicates is shown. **c** BG measurements of diabetic C57BL6/J mice following IP transplantation of SONIC devices (pink, $n = 5$, retrieved on day 60; red, $n = 6$, one device was retrieved on day 145 after BG rising and the other five were retrieved on day 181), or control devices (black, $n = 5$, two mice were sacrificed on day 96 and day 128 due to poor health, the other three devices were retrieved on day 181); ****$p < 0.0001$ (one-way analysis of covariance (ANCOVA)). **d** IPGTT on day 58; mean ± SD; ****$p < 0.0001$ (diabetic mice versus SONIC device-treated mice, diabetic mice versus healthy mice), n.s. ($p = 0.5591$; SONIC device-treated mice versus healthy mice). **e** IPGTT on day 180; mean ± SD; ****$p < 0.0001$ (diabetic mice versus SONIC device-treated mice, diabetic mice versus healthy mice, control device-treated mice versus SONIC device-treated mice, and control device-treated mice versus healthy mice), n.s. ($p = 0.6667$; diabetic mice versus control device-treated mice), n.s. ($p = 0.9966$; SONIC device-treated mice versus healthy mice). Statistical tests in d and e were analyzed via a two-way analysis of variance (ANOVA) followed by Sidak's post hoc $p$-value adjustment for multiple comparisons. **f** Static GSIS test of devices ($n = 3$) retrieved on day 60; mean ± SD, ***$p = 0.0006$ (paired two-sided students $t$-test). **g** Live/dead staining of islets from a retrieved SONIC device on day 60. One representative of 2 replicates is shown. **h, i** H&E and immunohistochemical staining of tangential cross-sections of retrieved devices on day 60 showing intact morphology and insulin/glucagon-positive islets in both peripheral regions (**h**) and central regions (**i**) of the device. One representative of 3 replicates is shown. **j** H&E staining of the transverse section of a retrieved device on day 181 showing healthy islets in both peripheral regions and central regions of the device. The asterisk indicates the host side of the device-host interface. One representative of 2 replicates is shown. **k** A selected transverse surface plot collected from the simulation device showing well-oxygenated islets in both peripheral regions and central regions. **l** H&E staining of a retrieved device on day 181 showing healthy islets in the device even with some deposited fibrosis on the device. The asterisk indicates the host side of the device-host interface. One representative of 4 replicates is shown. **m** Selected surface plots collected from the simulation of devices with fibrosis on one half of the device (implemented by a no-flux condition on the bottom half). The PLA control device (left) showed a significantly lower $pO_2$ at the blocked bottom side in comparison to the unblocked top side, and a necrotic islet (yellow arrow) was observed near the blocked face. The SONIC device (right) showed a slighter lower $pO_2$ at the blocked bottom side in comparison to the unblocked top side, but the islets (blue arrows) at both sides were sufficiently oxygenated.

We then fabricated a 4-layer scaffold using a new printed mold to test the model predictions (Supplementary Fig. 20). A 6.6 mm-thick SONIC device was prepared using the same procedure as described earlier (Fig. 2a). Rat islets (500 IEQ per transplant) were incorporated in devices and transplanted into the intraperitoneal cavity of diabetic C57BL/6 mice ($n = 10$), with 5 retrieved at day 60 and the remainder retrieved at day 120 (Fig. 6g, h). Strikingly, all mice achieved normoglycemia within a few days after transplantation, with 4 out of the 5 mice in the long-term study maintaining normoglycemia over 4 months (Fig. 6i). Additionally, the BG returned to hyperglycemia after device retrieval, confirming the role of the device in diabetes correction. An IPGTT performed on day 58 showed that the devices restored normoglycemia within 120 min, though the profile in device-treated mice showed a statistically significant delay in comparison to healthy control mice (Fig. 6j). BG monitoring was extended for an additional 1 h to monitor potential hypoglycemia; 6.6 mm SONIC treated mice showed a slight overcorrection to ~70 mg/dL at 150 min (compared to ~120 mg/dL in healthy control mice) but stabilized near this value at the 180 min time point. The observed response delay and hypoglycemic overcorrection is likely because of the significant diffusion distance for insulin secreted from deeply encapsulated islets. Nonetheless, subsequent live/dead staining further confirmed islet viability (Fig. 6k), Finally, H&E staining and insulin/glucagon immunostaining further demonstrated maintained viability and function of islets in the retrieved devices (Fig. 6l, m and Supplementary Fig. 21). This collective analysis indicates that the SONIC device was able to maintain islet survival and function in constructs significantly larger than traditional structures.

## Discussion

Herein, we report the design and testing of an insect-inspired scaffold (SONIC) which featured internal continuous air channels for rapid $O_2$ delivery to hydrogel encapsulated cells. Incorporation of this scaffold into bulk hydrogels containing islets (the SONIC device) supported high islet viability and robust function even in devices with a thickness of 6.6 mm. The SONIC scaffold was comprised of a hydrophobic fluoropolymer PVDF-HFP, and the internal continuous air channels were created by a phase separation process. A hydrophilic polydopamine coating was applied to the scaffold surface to provide a compatible interface

between the hydrophobic SONIC scaffold and hydrophilic hydrogel, while maintaining internal hydrophobicity to avoid water penetration into the air channels which were verified as essential for enabling the high $O_2$ permeability of the scaffold. Finally, the alginate hydrogel phase was interlocked with the skeleton of the SONIC scaffold to prevent hydrogel detachment during retrieval of the transplanted device.

$O_2$ supply is a primary limiting factor for cell encapsulation, especially for cells with high $O_2$ consumption rates such as islets. To overcome this limitation, much attention has been deservedly focused on exogenous $O_2$ supply, such as $O_2$-generating[15–18] and $O_2$-filling devices[13,14]. However, $O_2$ transport within the hydrogel-encapsulated cell component is driven by gradients of $O_2$ tension and remains slow due to the poor $O_2$ permeability in aqueous media. Fewer efforts have focused on improving $O_2$ transport within the cell encapsulation domain itself. Some reports have explored the effect of perfluorocarbon (PFC) emulsion incorporation in hydrogels due to the high $O_2$ solubility[56,57], and slightly higher diffusivity[58] in PFC compared to hydrogels. However, these systems generally only yield modest benefits because of the limited improvement in $O_2$ permeability in composite systems where the phase with the higher permeability (in this case, PFC) is dispersed[59]. The bicontinuous gas-phase, endowed with extremely high permeability, incorporated into the hydrogel by the SONIC scaffold facilitates the rapid permeation of $O_2$ throughout the device, thereby enormously improving the effective $O_2$ permeability of the system. Regardless of design, the current SONIC system is reliant on the $O_2$ available in the transplantation site (Supplementary Fig. 13); however, we also posit that the SONIC scaffold could be used to further enhance $O_2$ supply to hydrogel encapsulated cells even in devices that provide exogenous supply.

The mechanism of improved $O_2$ distribution via SONIC can be visualized accordingly: the continuous hydrogel phase of the device can be considered as comprising small cubic regions of $\sim 600 \times 600 \times 600$ μm (Supplementary Fig. 22). Each of these hypothetical "microcubes" are adjacent to the SONIC scaffold which contains uniform $pO_2$ levels comparable to those of the transplantation site. Thus, it is as if each hypothetical microcube itself was implanted individually in the intraperitoneal space. Furthermore, the interconnection of the hypothetical microcubes ensured that other water-phase nutrients (e.g., glucose), found in

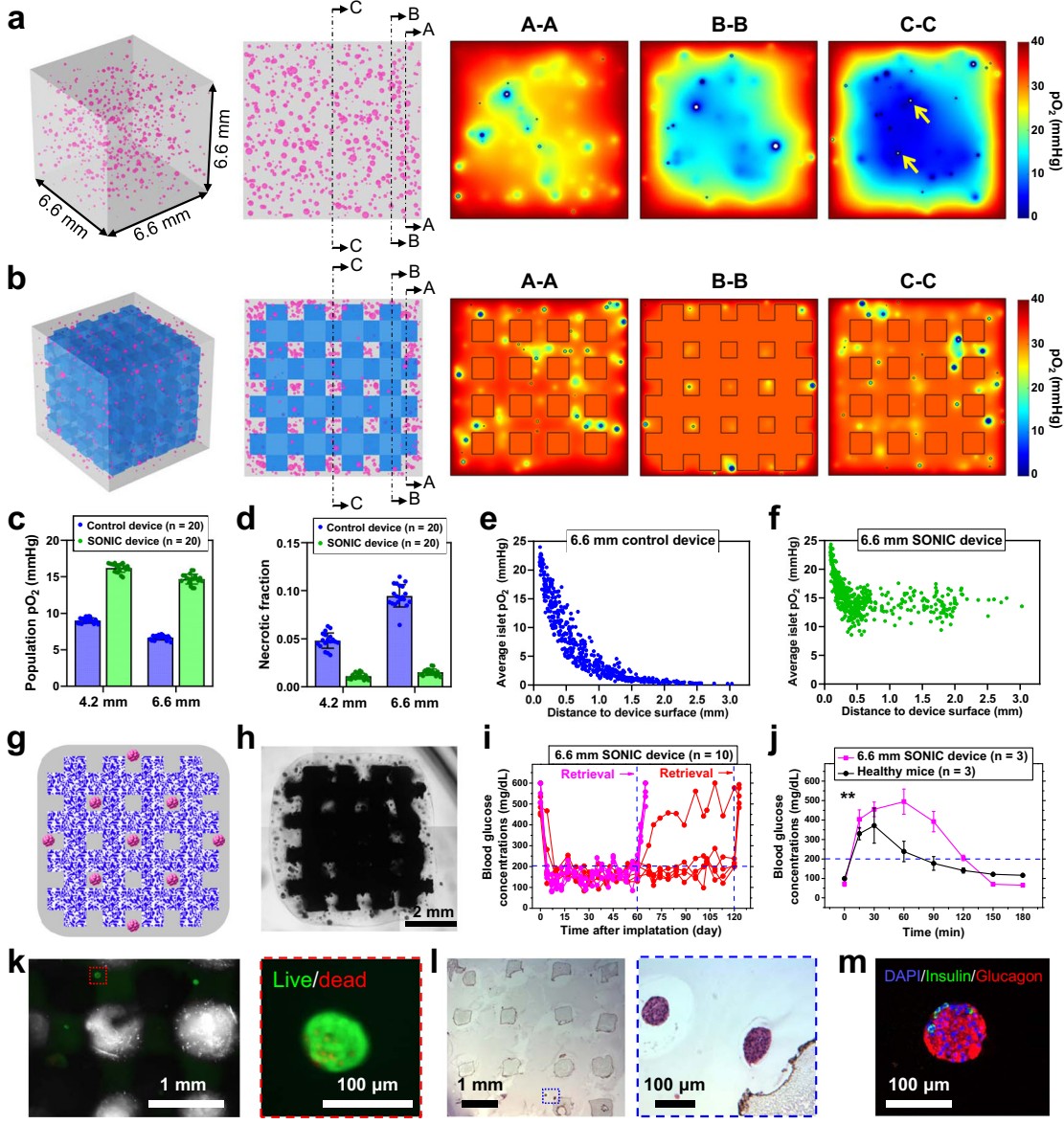

**Fig. 6 Demonstration of the therapeutic potential of a thick SONIC device. a, b** Schematics representing the simulated geometry for a thick control device
(**a**) and thick SONIC device (6.6 × 6.6 × 6.6 mm) (**b**) containing rat islets (size-distributed). Simulation-predicted pO₂ distributions of the control device in
three cross-sections (labeled A-A, B-B, and C-C) showing a massive hypoxic central region, with necrosis observed even in some small islets (arrows in
C-C cross-section). However, the islets were well-oxygenated throughout the SONIC device with negligible necrosis. **c, d** Average pO₂ (**c**) and the fraction
of necrosis (**d**) in the islet populations in control devices ($n = 20$) and SONIC devices ($n = 20$) at the size of 4.2 mm and 6.6 mm, mean ± SD. **e, f** Scatter
plots of islet location versus average pO₂ of islets (generated as all 150 μm in diameter) in the thick control device (**e**) and thick SONIC device (**f**).
**g, h** Schematic (**g**) and microscope image (**h**) of rat islets encapsulated in a thick cubic SONIC device with a side length of ~6.6 mm. One representative of
5 replicates is shown. **i** BG measurements of diabetic C57BL6/J mice following IP transplantation of the 6.6 mm SONIC devices (pink, $n = 5$, retrieved on
day 60; red, $n = 5$, retrieved on day 123). **j** IPGTT on day 58; mean ± SD; **$p = 0.0066$ (two-way ANOVA). **k** Live/dead staining of islets from a retrieved
device on day 120. **l, m** H&E (**l**) and immunohistochemical staining (**m**) of retrieved devices on day 120 showing morphology-intact and insulin/glucagon-
positive islets. One representative of 5 replicates is shown.

the host site in significantly higher concentrations than O₂[60],
permeate throughout the hydrogel phase.

A successful islet delivery implant must not only maintain cell
survival but also ensure timely release of insulin to prevent
postprandial hyperglycemia and overcorrection into hypoglyce-
mia. The delay in IPGTT observed in mice treated with the thick
SONIC device (Fig. 6j) indicate that, for islet delivery, the pos-
sibility of simply extending the device structure in all three
dimensions may be limited. However, cellular treatments for
hemophilia and liver diseases also utilize high cell payloads[5]
without the requirement of phasic therapeutic release. Therefore,

the thick SONIC system may find broader utility in these
applications.

We also emphasize that 3D printing enables the SONIC scaf-
fold to be fabricated in scaled-up dimensions (e.g., in multiple
layers or extended in length to tens of centimeters) in a wide
range of designs (e.g., toroidal, spiral) (Supplementary Fig. 23 and
Supplementary Fig. 24). An advantageous configuration of an
islet delivery device is a planar hydrogel disk (1.2 mm thickness)
incorporating an internal SONIC scaffold configured in an
Archimedean spiral with a 500 μm gap between turns (Fig. 7 and
Supplementary Fig. 25a). The small thickness of 1.2 mm obviates

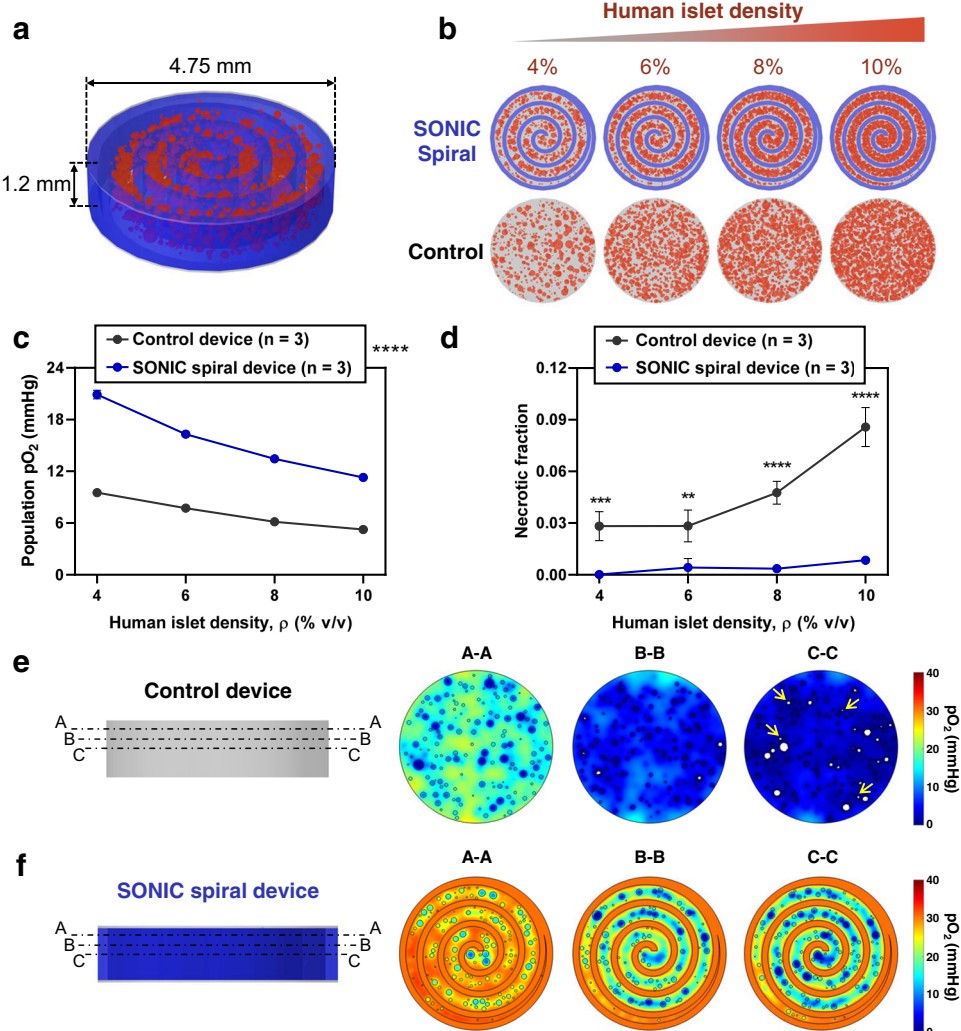

**Fig. 7 Computational exploration of a SONIC spiral device for delivering a clinically relevant islet dose. a** Annotated schematic of the central section of a hypothetical SONIC spiral device, including the SONIC scaffold (blue) arranged in an Archimedean spiral (with the distance between turns fixed at 500 μm) and hydrogel-encapsulated human islets (red). A thickness of 1.2 mm ensures a maximum distance of insulin diffusion of 600 μm; a diameter of 4.75 mm was used for simulations, representing the central section of a device scaled radially to achieve a sufficient encapsulated islet payload. **b** Schematics showing the SONIC spiral device and scaffold-free control device encapsulating 4%, 6%, 8%, and 10% human islets (volume of islets per volume of device), as tested in the simulations. **c, d** Simulation predictions of the mean islet population pO$_2$ (**c**) and fraction of necrosis (**d**) of human islets encapsulated at variable densities in the SONIC spiral device ($n = 3$) and the scaffold-free control device ($n = 3$); mean ± SD; c: ****$p < 0.0001$ (control device versus SONIC spiral device at all islet densities); d: ***$p = 0.0004$ (control device versus SONIC spiral device at 4% islet density), **$p = 0.0018$ (control device versus SONIC spiral device at 6%), ****$p < 0.0001$ (control device versus SONIC spiral device at 8% and 10% islet densities). Statistical tests in c and d were analyzed via a two-way ANOVA followed by Sidak's post hoc $p$-value adjustment for multiple comparisons. **e, f** Surface plots of pO$_2$ gradients (right) at selected cross-sections (left; labelled A-A, B-B, and C-C) in the control device (**e**) and the SONIC spiral device (**f**) at 8% human islet loading density showing significantly higher pO$_2$ and negligible necrosis in the SONIC spiral device compared to the control device (white regions in the islets represent necrosis, and yellow arrows in C-C section indicate necrosis even in some small islets).

limitations associated with delays in insulin release, and the spiral structure ensures that each islet is within 250 μm of a high pO$_2$ source in the SONIC scaffold (Fig. 7a). We assessed the capacity of this construct to accommodate variable densities (4–10%) of human islets in comparison to a control construct without the SONIC scaffold using the computational model (Fig. 7b). Model predictions indicate significantly higher oxygenation of islets in the SONIC spiral device relative to the control construct, and negligible necrosis levels up to 8% (v/v) human islet loading density (Fig. 7c–f). If extended radially, this construct could support a curative islet dose of 500 k IEQ within a disk approximately 11 cm in diameter.

In summary, this work provides a solution to the poor transport of O$_2$ in traditionally employed bulk hydrogels of cell encapsulation systems. SONIC's mimicry of the insect tracheal system yields a cell encapsulation device that is amenable to increased cell density, fibrotic blockage, and most notably, substantially increased device thickness without sacrificing cell oxygenation, in effect, decoupling cell survival from its distance to the external supply. Ultimately, these advantages imparted by the SONIC scaffold represent a promising platform for translatable encapsulation devices requiring high cell payloads.

## Methods

**Materials**. Poly(vinylidene fluoride-*co*-hexafluoropropylene) (PVDF-HFP, Mw = 455 kDa), Tris hydrochloride (Tris-HCl), sodium hydroxide, dopamine hydrochloride, sodium chloride (NaCl), calcium chloride dihydrate (CaCl$_2$·2H$_2$O), barium chloride dihydrate (BaCl$_2$·2H$_2$O), calcium sulfate dihydrate (CaSO$_4$·2H$_2$O),

Nile Red, gelatin and *D*-glucose were purchased from Sigma-Aldrich. Poly(lactic acid) (PLA) filament was purchased from PRUSA. Ultrapure sodium alginate (Pronova SLG100) was purchased from NovaMatrix. Water was deionized to 18.2 MΩ cm with a Synergy UV purification system (Millipore Sigma).

**Animals**. Male C57BL/6 J mice (2 months old) were purchased from The Jackson Laboratory. The mice were maintained at a temperature of 70–72 °F with 30–70% humidity under a 14 h light/10 h dark cycle. Male Sprague-Dawley rats (weight of ~300 g) were purchased from Charles River Laboratories. All animal procedures were approved by the Cornell Institutional Animal Care and Use Committee and complied with relevant ethical regulations.

**Characterizations**. High-resolution X-ray computer tomography (Nano-CT) scanning was conducted on a 3D X-ray microscope (ZEISS Xradia 520 Versa). Scanning electron microscopy (SEM) and energy-dispersive X-ray spectroscopy (EDS) element mapping were performed using a field emission scanning electron micro-analyzer (LEO 1550). Contact angle images were taken using a contact angle goniometer (Rame-Hart 500). Optical and fluorescent microscope images were taken using a digital microscope (EVOS FL). H&E staining images were taken using an Aperio Scanscope (CS2). Stereo microscope images were taken by a stereomicroscope (Olympus SZ61). Immunofluorescence images were taken using a confocal microscope (ZEISS LSM 710). OriginPro 8.5.1 software and GraphPad Prism 8 software were used for data plotting.

**Fabrication of the SONIC scaffold**. PVDF-HFP was dissolved in acetone at a concentration of 15 wt% under heat in a sealed glass vial. After cooling to room temperature, the PVDF-HFP solution was filled into a 3D printed PLA mold (Original Prusa i3 MK2S) and immersed in a water/ethanol (V/V = 1/1) bath for a phase separation process, and then transferred to a water bath for a solidification process. Next, the solidified PVDF-HFP was immersed in ethanol and hexane for two dehydration processes, followed by air drying at ambient temperature. Finally, the SONIC scaffold was obtained after the selective extraction of the PLA mold with chloroform. (The SONIC scaffolds for in vivo studies were sterilized by autoclave.)

**Nano-CT imaging of the mealworm and SONIC scaffold**. To prepare a mealworm specimen for Nano-CT scanning, a 2 cm-long mealworm was loaded in a 1 mL pipet tip (Supplementary Fig. 1) and sacrificed by freezing at −20 °C. 6 individual scans were performed on different sections of the mealworm using an "oversize scan" option to get a full image of the specimen. During the scans, the X-ray source was set to a voltage of 100 kV, and the scanning resolution was set as 5.19 μm per pixel under a binning mode of 2 × 2. Subsequently, a 3D reconstruction of the obtained images was performed using Avizo software (version 8.1.1). A segmentation process was conducted to visualize the tracheal system of the mealworm based on the different absorption contrasts between the respiratory gases and mealworm tissues.

To prepare a SONIC scaffold specimen for the Nano-CT scanning, a small piece of scaffold (~1 mm³) was cut and attached on a tip. During the scan, the X-ray source was set to a voltage of 100 kV, and the scanning resolution was set as 0.268 μm per pixel under a binning mode of 2 × 2. Subsequently, 3D reconstruction of the obtained images was performed using Avizo software. Network connectivity on the polymeric and the porous regions of the scaffold was performed using ImageJ.

**Electron paramagnetic resonance (EPR) for O₂ mapping**. O₂ mapping was performed on a 25 mT EPR imager (JIVA-25, O2M Technologies, LLC). The JIVA-25 operates at 720 MHz using electron paramagnetic resonance oxygen imaging (EPROI) principles and utilizes oxygen sensitive electron spin-lattice relaxation rates ($T_1$) of trityl radical probe OX063-D24 (methyl-tris[8-carboxy-2,2,6,6-tetra-kis[(2-hydroxyethyl)benzo[1,2-d:4,5-d']bis[1,3]dithiol-4-yl]- trisodium salt) for reporting pO₂.

A SONIC scaffold or control scaffold was fixed at the bottom of a glass tube (VWR, 10 × 75 mm) using a dental vinyl polysiloxane impression material. 1 mL gelatin solution (1 wt%) containing spin probe (1 mM OX063-d24) was filled into the container with the top end of the scaffold exposed above the gelatin. First, the system was deoxygenated using N₂ to reduce pO₂ close to 0 mmHg. After deoxygenation, the system was exposed to a gas mixture containing 5% O₂ and 95% N₂, and the pO₂ change in gelatin was continuously monitored until a steady-state (40 mmHg) was approached.

Average pO₂ measurements were performed using inversion recovery electron spin-echo (IRESE)[61] sequence with the following parameters: pulse lengths 60 ns, 16 phase cycles scheme with FID suppression, spin-echo delay 500 ns, 80 logarithmically spaced delays from 400 ns–65 μs, 100 us repetition time. The curves were fitted using single exponential recovery to extract $R_1$ ($1/T_1$) values that were converted to pO₂ (Fig. 3c). pO₂ imaging was performed using IRESE sequence with the following parameters: pulse lengths 60 ns, 16 phase cycles scheme with FID suppression, spin-echo delay 400 ns, equal solid angle spaced 654 projections, 67 baselines, 1.5 G/cm gradient, 10 time delays from 410 ns–40 μs, 35 μs–59 μs repetition time, overall 10 min image duration. Images were reconstructed using

filtered back-projection in isotropic 64 × 64 × 64 cube with 0.66 mm voxel linear size.

**Fabrication of the SONIC device**. The SONIC scaffold was immersed into a dopamine solution (2 mg/mL in 10 mM tris buffer, pH 8.5) overnight to create a hydrophilic polydopamine coating on the scaffold surface. Subsequently, CaSO₄ was deposited onto the scaffold surface by dipping it into a CaSO₄ saturated solution (0.24 wt% in water) and then drying it at 60 °C, leaving CaSO₄ crystals on the scaffold surface. Next, the scaffold was inserted into a glass tubing mold with sodium alginate (2%) solution. Alginate cross-linking then occurred by the Ca²⁺ ions diffused from the CaSO₄. Finally, the SONIC device was pushed out from the tubing mold into a cross-linking buffer (95 mM CaCl₂ + 5 mM BaCl₂), leaving the device in around 4 min for further cross-linking. Constructs that contained INS-1 cells or islets were fabricated by premixing the alginate solution with the cells before application onto the scaffolds.

To fabricate control devices without the SONIC scaffold, a tubing mold was prepared by rolling a dialysis membrane (Spectra/Por®, MWCO 3500) into a tube with an inner diameter of ~4 mm and sealing one end with a PDMS cap. Then, INS-1 cells or islets alginate solution were loaded into the mold and immersed in the buffer (95 mM CaCl₂ + 5 mM BaCl₂) for cross-linking by the Ca²⁺ and Ba²⁺ ions which diffused through the dialysis membrane. Next, the tubing mold was unrolled to leave the alginate in the buffer for around 4 min for further cross-linking.

For the devices in mice studies, 500 IEQ of rat islets distributed in approximately 170 μL alginate were incorporated in each cylindrical (4.2 mm in diameter, 20.4 mm in length) SONIC device (Fig. 5b) and 500 IEQ of rat islets distributed in approximately 280 μL alginate were incorporated in each corresponding cylindrical scaffold-free control device (Supplementary Fig. 14); 500 IEQ of rat islets distributed in approximately 160 μL alginate were incorporated in each cubic (6.6 × 6.6 × 6.6 mm) SONIC device (Fig. 6h).

**In vitro cell viability study**. INS-1 cells were purchased from Sigma-Aldrich and cultured with RPMI 1640 medium (Gibco) supplemented with 10% FBS (Gibco), 10 mM HEPES (Gibco), 2 mM glutamine (Gibco), 1 mM sodium pyruvate (Gibco), 50 μM β-mercaptoethanol (Gibco), and 1% penicillin/streptomycin (Gibco). Trypsin-dissociated INS-1 cells were suspended in alginate solution at a density of 2.5 million cells/mL and incorporated into SONIC devices or control devices, and then were incubated in the above-mentioned medium in a hypoxic incubator with 5% O₂, 5% CO₂ at 37 °C. After 48 h, the cells in devices were stained with a LIVE/DEAD™ viability/cytotoxicity kit (Invitrogen).

**Rat islet isolation and purification**. Sprague-Dawley rats were used for harvesting islets. The rats were anesthetized using 3% isoflurane in O₂ throughout the whole surgery. Briefly, the pancreas was distended with 10 mL 0.15% Liberase (Roche) in M199 media (Gibco) through the bile duct. The pancreas was digested at 37 °C circulating water bath for ~28 min (digestion time varied slightly for different batches of Liberase). The digestion was stopped by adding cold M199 media with 10% FBS (Gibco). After vigorously shaking, the digested pancreases were washed twice with media (M199 + 10% FBS), filtered through a 450 μm sieve, and then suspended in a Histopaque 1077 (Sigma)/M199 media gradient and centrifuged at 1700 RCF with 0 breaks and 0 acceleration for 17 min at 4 °C. This gradient centrifugation step was repeated for higher purity. Finally, the islets were collected from the gradient and further isolated by a series of gravity sedimentations, in which each top supernatant was discarded after 4 min of settling. Islet equivalent (IEQ) number of purified islets was counted by reported IEQ conversion factors[62]. Islets were then washed once with islet culture media (RPMI 1640 supplemented with 10% FBS, 10 mM HEPES, and 1% penicillin/streptomycin) and cultured in this medium overnight before further use.

**Implantation and retrieval in mice**. C57BL/6 J mice were administered an intraperitoneal injection of freshly prepared STZ solution (22.5 mg/mL in 100 mM sodium citrate buffer, pH 4.5) at a dosage of 150 mg STZ/kg mouse to induce diabetes one week before device implantation. Only mice with non-fasted blood glucose levels above 350 mg/dL were considered as diabetic. The diabetic mice were anesthetized with 3% isoflurane in O₂ and the abdomen aera was shaved and sterilized using betadine and 70% ethanol. A small skin incision (~8 mm) was made along the midline of the abdomen, and then a following incision was made along the linea alba. The device was introduced into the peritoneal cavity through the incision. The peritoneal wall was closed using 5-0 absorbable polydioxanone (PDS II) sutures and the skin incision was closed using 5-0 nylon sutures.

For retrieval, the mice were treated with the same procedures as above. Then, the device was located and pulled out from the peritoneal cavity using a tweezer. The incisions were sutured and keep the mice alive for following BG monitoring after device retrieval.

**Morphology and immunohistochemistry of islets in retrieved devices**. The retrieved devices were fixed with 10% formalin, embedded in paraffin, and sectioned into 5 μm sections. Hematoxylin and eosin (H&E) staining was performed by Cornell's Histology Core Facility. For immunofluorescent insulin and glucagon

staining, paraffin-embedded sections were deparaffinized in xylene and sequentially rehydrated in 100% ethanol, 95% ethanol, 75% ethanol, and PBS. Slides were then boiled in citric acid buffer (10 mM citric acid, 0.05% Tween 20, pH 6.0) for 30 min for antigen retrieval. After blocking with 5% donkey serum, primary rabbit anti-rat insulin (Abcam, ab63820, 1:200) and mouse anti-rat glucagon (Abcam, ab10988, 1:200) antibodies were applied and incubated overnight at 4 °C. After washing with PBS, Alexa Fluor 594-conjugated goat anti-rabbit IgG (Thermofisher, A11037, 1:400) and Alexa Fluor 488-conjated donkey anti-mouse IgG (Thermofisher, A21202, 1:400) were applied and incubated for 60 min. Finally, slides were washed with PBS, applied with antifade/DAPI, and covered with glass coverslips.

**BG monitoring & intraperitoneal glucose tolerance test (IPGTT).** Mouse BG levels were measured by a commercial glucometer (Contour Next EZ, Bayer) with a drop of blood collected from the tail vein. For the IPGTT, mice were fasted for 16 h and administered an intraperitoneal injection of 20% glucose solution at a dosage of 2 g glucose/kg mouse. BG levels were measured at 0, 15, 30, 60, 90, and 120 min following the injection.

**Ex vivo static glucose-stimulated insulin secretion (GSIS) assay.** Krebs Ringer Bicarbonate (KRB) buffer was prepared according to the following formula: 98.5 mM NaCl, 4.9 mM KCl, 2.6 mM CaCl$_2$·2H$_2$O, 1.2 mM MgSO$_4$·7H$_2$O, 1.2 mM KH$_2$PO$_4$, 25.9 mM NaHCO$_3$, 0.1% BSA (all from Sigma-Aldrich), and 20 mM HEPES (Gibco). The retrieved devices were incubated in the KRB buffer for 2 h at 37 °C, 5% CO$_2$. Devices were transferred and incubated in KRB buffer supplemented with 3.3 mM glucose, then 16.7 mM glucose for 75 min each. The buffer was collected after each incubation step, and insulin concentration was measured using an ultrasensitive rat insulin ELISA kit (ALPCO).

**Computational modeling.** Five general finite element models were created to calculate theoretical O$_2$ profiles in SONIC-enabled constructs and corresponding controls. In all models, O$_2$ tension (pO$_2$) was related to the concentration of O$_2$ (c$_{O_2}$) by the Bunsen solubility, or equilibrium concentration of O$_2$ in a material i at 37 °C, α$_{O_2,i}$:

$$c_{O_2} = \alpha_{O_2,i} \cdot pO_2 \qquad (1)$$

In other words, O$_2$ partitioning was governed by Henry's law where, in Eq. 1, α$_{O_2,i}$ represents the inverse of Henry's constant. Each model is described below.

Model 1 (Fig. 3h, i and Supplementary Fig. 5) simulated time-dependent oxygenation of the in vitro acellular test. Two domains were considered: a rectangular prism representing the SONIC scaffold or PLA control (2 × 2 × 23 mm), and a surrounding cylinder (8.6 mm diameter, 17 mm length), representing gelatin, with the scaffold located in the center of the gelatin (Supplementary Fig. 5a, b). To emulate the experimental set up, boundary conditions at the top face of the gelatin and exposed faces of the scaffold were implemented at a constant pO$_2$ of 40 mmHg, while no flux conditions were implemented on all other faces due to the O$_2$ impermeability of the glass test tube and fixing resin containing the system (Supplementary Fig. 5c). Transient pO$_2$ transport in each domain was governed by Fick's second law:

$$\alpha_{O_2,i}\frac{\partial(pO_2)}{\partial t} = -\alpha_{O_2,i}D_{O_2,i}\left(\frac{\partial^2(pO_2)}{\partial x^2} + \frac{\partial^2(pO_2)}{\partial y^2} + \frac{\partial^2(pO_2)}{\partial z^2}\right) \qquad (2)$$

here, D$_{O_2,i}$ represents diffusivity of O$_2$ in domain i (i.e., SONIC scaffold, PLA, or gelatin) at 37 °C. As it was confirmed that a bicontinuous porous structure was maintained throughout the SONIC scaffold (Fig. 1j, k, Supplementary Fig. 3, and Supplementary Movie 2), this domain was modeled as a gaseous air phase. Therefore, α$_{O_2,SONIC}$ = 3.9 × 10$^{-4}$ mol/(m$^3$ Pa) rather represents the equilibrium O$_2$ concentration in air and was calculated by the ideal gas law. Solubility and diffusivity values for gelatin varied widely in the literature, thus values for alginate hydrogel were used, given their physical and chemical similarity. The remaining solubilities were thus implemented as α$_{O_2,gelatin}$ = α$_{O_2,alginate}$ = 9.3 × 10$^{-6}$ mol/(m$^3$ Pa)[59], and α$_{O_2,PLA}$ = 4.5 × 10$^{-5}$ mol/(m$^3$ Pa)[63], each obtained from the literature. Likewise, D$_{O_2,SONIC}$ = 1.8 × 10$^{-5}$ m$^2$/s[64], D$_{O_2,gelatin}$ = D$_{O_2,alginate}$ = 2.7 × 10$^{-9}$ m$^2$/s[59,65], D$_{O_2,PLA}$ = 1.6 × 10$^{-12}$ m$^2$/s[63], also obtained from the literature, were implemented for O$_2$ diffusivities in the respective materials. A time-dependent study simulated over 80 h was performed with a time step of 5 mins.

Model 2 simulated steady-state O$_2$ transport in cylindrical constructs (4.2 mm diameter, 20.4 mm length) containing alginate-encapsulated INS-1 cells with cell densities from 1.0–8.0 million cells per mL alginate (Fig. 4a–d and Supplementary Fig. 7). O$_2$ profiles in a construct featuring the ladder-like SONIC scaffold were compared to those in a scaffold-free control (Supplementary Fig. 7a). The alginate and encapsulated cells were modeled as one composite domain, with diffusivity and solubility coefficients of that of alginate. A constant pO$_2$ of 40 mmHg was implemented at all external boundaries (Supplementary Fig. 7b). Steady-state pO$_2$

profiles were obtained by solving the diffusion-reaction mass balance equation:

$$\alpha_{O_2,i}D_{O_2,i}\left(\frac{\partial^2(pO_2)}{\partial x^2} + \frac{\partial^2(pO_2)}{\partial y^2} + \frac{\partial^2(pO_2)}{\partial z^2}\right) = -R_{O_2} \qquad (3)$$

Above, R$_{O_2}$ represents O$_2$ consumption by the encapsulated INS-1 cells, modeled using Michaelis-Menten kinetics and a step-down function[66]:

$$R_{O_2}(pO_2) = \begin{cases} 0, pO_2 < 0.08\text{mmHg} \\ \frac{V_{INS-1}\cdot\rho}{\alpha_{O_2,alginate}}\cdot\left(\frac{pO_2}{pO_2+K_m}\right), pO_2 \geq 0.08\text{mmHg} \end{cases} \qquad (4)$$

In Eq. 4, V$_{INS-1}$ = 5.0 × 10$^{-17}$ mol/(m$^3$ s cell) represents the literature-retrieved INS-1 cellular O$_2$ consumption rate[67], K$_m$ = 0.81 mmHg represents the half-maximum constant derived from studies on mitochondrial respiration[68], and ρ represents the cell density which was implemented at 2.5 million cells/mL to match experimental conditions (Fig. 4a–f) but also varied between 1–8 million cells/mL to explore the effect of varying cell density (Supplementary Fig. 7d, e). Below the threshold of pO$_2$ = 0.08mmHg, O$_2$ consumption is set to zero[48,69], representing the lack of respiration in necrotic cells as in models described elsewhere.

Model 3 simulated steady-state O$_2$ transport in all cylindrical constructs (4.2 mm in diameter, 20.4 mm or 6.4 mm in length) containing alginate-encapsulated rat islets (Fig. 4g–l, Fig. 5k, m, Supplementary Fig. 8–11, Supplementary Fig. 13, and Supplementary Fig. 18; 500 IEQ rat islets per device) or human islets with cell densities from 2.74–8.04% v/v in devices (4.2 mm in diameter, 2.2 mm in length) (Supplementary Fig. 12). In all cases, islets were implemented as perfect spheres and seeded randomly in the alginate domain. As a default, islet diameters, d, were randomly selected from a size distribution. Rat islet diameters were selected from a lognormal distribution (Supplementary Fig. 8b), with the probability density function given by:

$$f(d) = \frac{1}{d\alpha\sqrt{2\pi}}\exp\left(-\frac{\ln(d/\beta)^2}{2\alpha^2}\right) \qquad (5)$$

where α = 0.40 and β = 112.6 (these values were obtained empirically and were found to be similar to distributions observed in other animal islet sources)[70]. Human islet diameters were selected from Weibull distribution[62] (Supplementary Fig. 12c), with the probability density function given by:

$$f(d) = \frac{\alpha}{\beta}\left(\frac{d}{\beta}\right)^{\alpha-1}\exp\left(-\left(\frac{d}{\beta}\right)^{\alpha}\right) \qquad (6)$$

where α = 1.5 and β = 105. In specified cases (Fig. 4k, l and Supplementary Fig. 10), islets were instead all generated with d = 150 μm.

Steady-state pO$_2$ profiles were obtained by solving the diffusion-reaction mass balance equation (Eq. 3). Solubility and diffusivity in the islets were given by α$_{O_2,islets}$ = 7.3 × 10$^{-6}$ mol/(m$^3$ Pa) and D$_{O_2,islets}$ = 2.0 × 10$^{-9}$ m$^2$/s respectively[57,64,65]. In this model, R$_{O_2,i}$ was only defined in the islet domains according to the following:

$$R_{O_2}(pO_2) = \begin{cases} 0, pO_2 < 0.08\text{mmHg} \\ \frac{V_{islets}}{\alpha_{O_2,islets}}\cdot\left(\frac{pO_2}{pO_2+K_m}\right), pO_2 \geq 0.08\text{mmHg} \end{cases} \qquad (7)$$

where V$_{islets}$ = 0.0340 mol/(m$^3$ s) represents the O$_2$ consumption rate in rat islets[71] and V$_{islets}$ = 0.0134 mol/(m$^3$ s) in human islets[50], respectively. Model 3 was used to predict islet oxygenation and necrosis in cylindrical constructs implanted intraperitoneally in mice (Fig. 4g–j), whereby a constant pO$_2$ of 40 mmHg was implemented on all external boundaries (Supplementary Fig. 8c). The effect of variable external boundary pO$_2$ was explored in Supplementary Fig. 13. This model was also used to test the hypothetical impact of alternative scaffold compositions, including PLA, solid PVDF-HFP, or porous PVDF-HFP/alginate in constructs (Supplementary Fig. 8d, e). Solubility and diffusivity coefficients[72,73] for the alternative scaffold materials are listed in Table S1. Model 3 was also used to predict the influence of partial fibrosis in the same constructs (Fig. 5k, m and Supplementary Fig. 18), modeled by a no flux condition implemented on one half of the exterior of a cylindrical device (SONIC versus scaffold-free control, each 20.4 mm length) containing 500 IEQ rat islets to imitate a severe case of blockage by a fibrotic layer (Fig S17a). Oxygenation of 500 IEQ rat islets in cylindrical constructs of variable lengths (6.4 mm vs. 20.4 mm) and therefore cell densities was also evaluated with this model (Supplementary Fig. 11). Finally, Model 3 was used to evaluate human islet oxygenation and necrosis in cylindrical devices (SONIC versus scaffold-free) at variable densities (Supplementary Fig. 12).

Model 4 simulated steady-state O$_2$ transport in two thick cubics (6.6 × 6.6 × 6.6 mm) devices, each containing 500 IEQ rat islets (Fig. 6a–f, and Supplementary Fig. 19): a scaffold-free control device and a SONIC device. A constant pO$_2$ of 40 mmHg was applied to all exterior boundaries. All physics implementations of Model 4 were identical to those of Model 3, except for the dimensions, which are defined in Supplementary Fig. 19a, b.

Model 5 simulated steady-state O$_2$ transport in the SONIC spiral device and empty control containing variable loading densities of human islets. A constant pO$_2$ of 40 mmHg was imposed on the top and bottom boundaries whereas a no-

flux condition was imposed on the lateral face, as the modeled geometry is intended to represent only the central region of a device which would be extruded radially. All other physics implementations were identical to those of Model 3, except for the dimensions, which are defined in Supplementary Fig. 25a.

All models were solved in COMSOL Multiphysics or COMSOL Livelink for MATLAB. In Models 3–5, all calculations were repeated for at least 3 iterations, whereby the islets were reselected and repositioned at random each time. For all calculations, a mesh was implemented using COMSOL's "Free Tetrahedral" program with the following settings: maximum element size of 100 µm, minimum element size of 1 µm, curvature factor of 0.3, resolution of narrow domains of 3.3, and maximum growth rate of 1.25. It was ensured that all results were independent of the mesh.

**Statistics**. All results are expressed as raw data or as mean ± SD. Data from random BG measurements (Fig. 5c) were analyzed via a one-way analysis of covariance (ANCOVA) where device treatment (e.g., control device or SONIC device) was considered a discrete factor and time was considered a continuous covariate. Here, data from two-month SONIC device-treated mice (pink) were compared to data from control-treated mice (black) between days 4 and 61, while data from 6-month SONIC device-treated mice (red) were compared to data from control-treated mice (black) between days 4 and 181. Data from IPGTT studies (Fig. 5d, e and Fig. 6j) were analyzed via two-way analysis of variance (ANOVA) where both time and treatment (e.g., diabetic control mice, healthy control mice, control device-treated mice, and SONIC device-treated mice) were considered discrete factors, followed by a Sidak's post hoc $p$-value adjustment for multiple comparisons. Data from the GSIS test (Fig. 5f) was analyzed via a paired two-tailed students $t$-test. Average population $pO_2$ and necrotic fraction data from modeling studies (Fig. 4i, j) were analyzed via an unpaired two-sided students $t$-test. Data from modeling studies of the thick hydrogel (Fig. 6c, d), variable scaffold compositions (Supplementary Fig. 11d, e), variable density studies (Fig. 7c, d and Supplementary Fig. 12), and variable external $pO_2$ (Supplementary Fig. 13) were analyzed via a two-way ANOVA followed by Sidak's post hoc $p$-value adjustment for multiple comparisons. Statistical significance was concluded at $p < 0.05$.

**Reporting summary**. Further information on research design is available in the Nature Research Reporting Summary linked to this article.

## Data availability
All data supporting the findings of this study are available within the article and the Supplementary Information files and from the corresponding author upon reasonable request. Source data are provided with this paper.

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

## Acknowledgements

This work was partially supported by the National Institutes of Health (NIH, 1R01DK105967-01A1), the Novo Nordisk Company, the Juvenile Diabetes Research Foundation (JDRF, 2-SRA-2018-472-S-B), and the Hartwell Foundation. The work made use of the Cornell Center for Materials Research Shared Facilities which are supported through the NSF MRSEC program (DMR-1719875). This material is also based upon work supported by the National Science Foundation Graduate Research Fellowship under grant number DGE-1650441. We thank Cornell University Animal Health Diagnostic Center for histological sectioning and staining. We thank the BRC Imaging Facility at Cornell's Institute of Biotechnology for Nano-CT analysis (NIH S10OD012287). O2M Technologies acknowledges the support of JDRF grant 3-SRA-2020-883-M-B, NIH/NCI SBIR grants R43CA224840, R44CA224840, and NSF SBIR grant 1819583.

## Author contributions

L.-H.W. and M.M. conceived and designed the experiments. L.-H.W. and D.A. performed the experiments. L.-H.W. and A.U.E performed the data analysis. A.U.E and A.K.D. performed the computational modeling. B.E. and M.K. contributed to the EPR O2 imaging. L.-H.W., A.U.E., and M.M. wrote the manuscript.

## Competing interests

B.E. discloses financial interests in O2M Technologies. L.-H.W., A.U.E., and M.M. are inventors on a patent (no. US 63/174,739) based on this work filed by Cornell University on April 14, 2021. All other authors declare no competing interests.

## Additional information

**Peer review information** *Nature Communications* thanks Clark Colton and the other anonymous reviewer(s) of this work. Peer reviewer reports are available.

