## [Peer Review File · Nature Communications]

Reviewers' Comments:

Reviewer #1:

Remarks to the Author:

This is an impressive study performed by Wang et al, who have designed a material to facilitate greater O₂ transport to cells embedded in hydrogels. Following a bio-inspired design, the authors have designed a material to enable O₂ transport in the gas phase throughout bicontinuous, non-wettable micro-channels in PVDF-HFP scaffolds. The material does not generate oxygen. Rather, it enables oxygen transport from the ambient surroundings to cells encapsulated in hydrogel that is solidified around the scaffold. Using this strategy, the authors have demonstrated that they can provide a more spatially uniform oxygen tension throughout the hydrogel and improve cell viability and long-term in vivo efficacy.

In the study, the authors first characterized the material structure, wettability properties, and oxygen transport properties. They demonstrated that the oxygen transport properties were superior to that of a PLA scaffold control. The authors then compared the viability of cells encapsulated in alginate hydrogel around their scaffold to that of cells in alginate hydrogels only and showed that the presence of the hydrogel increased cell viability. Similarly, they showed that alginate hydrogels encapsulating islet cells and implanted in vivo without their scaffolds were not able to treat diabetic mice for extended periods, whereas those with their scaffolds could treat diabetic mice for up to 6 months (and it was reversed when the scaffold was removed). Overall, the authors present robust characterization that demonstrates their scaffold enhances oxygen transport from the surroundings and promotes the therapeutic benefit of islet cells .

One critique of the study is the controls chosen. The solution to oxygen transport is premised upon the non-wettability of the continuous microchannels (that facilitated gas phase O₂ transport) combined with the wettability of the scaffold exterior that enabled hydrogel encapsulation around the material. The controls for the study were not chosen to demonstrate that it is this particular combination of features that provides the advantage. The controls used were either solid PLA scaffold or no scaffold. It would be more convincing to demonstrate that it is not simply the microporous polymer that provides the benefit. Specifically, there is no ambient gas phase O₂ in the peritoneum, therefore the presumption is that O₂ must come out of solution to be transported in the gas phase to encapsulated cells.

Reviewer #2:

Remarks to the Author:

Review of NCOMMS-20-4960

A bioinspired scaffold for rapid oxygenation of cell encapsulation systems

This study describes a novel approach for cell therapy of diabetes by using a porous scaffold (akin to closely connected segments of porous tubes) to bring air in close proximity in situ to islets encapsulated in a gel. The oxygen diffuses through the gas in the porous space of the scaffold. The air is supplied through external surfaces in contact with tissue or fluid in the peritoneal space. The method described is a significant variation of the notion of delivering oxygen in situ to encapsulated cells, rather than solely through the adjacent microvasculature, which was described originally more than 20 years ago and manifested more recently in designs that utilize islets encapsulated in a planar

configuration exposed to elevated oxygen supplied exogenously. The main differences are the way in which the oxygen is supplied from the surrounding tissue through the porous structure instead of pumping oxygen into the device from an implanted or external source.

The paper gives convincing evidence that adequate, rapid oxygenation over the entire domain of islets is achieved in the experiments carried out. The methods are comprehensive and the results mostly convincing. However, the approach suffers from flaws that render it impractical for actual use, and the configuration used in some of the experiments is likely not representative of what is achievable in a clinical device.

Major Issues

In proposing a new device concept for therapeutic use, it is essential to consider the ultimate utility in for the design of a clinical device and the limitations and potential problems that might occur in achieving required function of the device, which is not done in the manuscript. In addition, the resultsof this study should be compared to that of other approaches in the literature that use in situ supply of oxygen.

1. The literature review is incomplete. Previous studies with configurations in which O₂ supply is enhanced and not limited to passive diffusion from surrounding blood vessel is barely mentioned. This study should be placed in the context of and compared to, other studies using in situ supply of oxygen, in particular those that supply (elevated) oxygen within the device to cells encapsulated in a planar configuration. Examples include Canadian Patent Application 2924681 and refs 101and 107cited in Ref 14 of this manuscript.

2.. The maximum islet concentration in the gel regions examined, based on data provided for both islets and cells, is estimated to be less than 0.01 (v/v). On a macroscopic basis (including the entire volume), the islet density is exceedingly low (volume fraction of about 0.003). An islet dose of 500,000 islets requires a volume of about 0.89 ml leading to a total device volume of about 300 ml. A clinical device could be arrayed in a variety of ways by varying shape, dimensions, and totalsurface area, leading to large length scales (thickness), and/or very large area, but there is no conceivable effective arrangement that would not be a surgical nightmare to implant this large volume for a clinical device in one or more modules in the peritoneum or in any body space for thatmatter.

3. There is no provision for an immunobarrier of any kind. This greatly reduces the value of using an encapsulation device because immunosuppression is still required for both allogenic and xenogenic cell sources. Provision of such a necessary barrier would add additional diffusive resistance for bothoxygen and insulin.

4. Because of the large total volume and the large fraction of volume devoted to the insulin-impermeable porous oxygen supply scaffold, the diffusion path for secreted insulin is extremely large for any configuration. The successful oxygenation of islets kept far from vessels produces a major flaw for dynamic supply of insulin to the bloodstream.

The secreted insulin must diffuse sufficiently rapidly through the device in order to achieve glucose homeostasis Hypohlycemia is a potential result of too slow response, and it can lead to coma and death. Estimates in the literature suggest a time constant for the response to a step change in BG should be on the order of 15 minutes or less in humans, which likely cannot be met with this configuration. The time constant for the scaffold device for the simplest case of Model 4 with a halftickness

of 0.33 cm and an insulin diffusivity of about 1×10^{-6} cm²/s is $t \sim L^2/D = 100,000$ s (about 30 h), which indicates that the insulin secreted from the islets at the center of the device would take an exceedingly long time to reach the bloodstream no matter what specific design is used, leading to a very dangerous situation with a device that could not fulfill its primary function.

Figures 5d and 5e show mouse IPGTT responses that are not normal but are closer to normal than to control devices (without scaffold), which at first suggests that the dynamics of insulin secretion are not terribly abnormal with the device. However, these results are misleading because they were obtained with the thin device that contains only three layers of islets. (Alternatively, the specific small device used can be viewed macroscopically as a cylindrical geometry.) When viewed as a

layered device, the top and bottom layers sit immediately adjacent to the device exterior (host tissue), which means that the insulin from these islets needs to diffuse only a small distance (to nearest blood vessels). Thus, the near-normal IPGTT results from the rapid response of the insulin from 2/3 of the islets in the device (if the structure is a square.). In the thicker device with 5 layers of islet in a planar configuration, the surface layers constitute only 40% of the islet contents, so the response would be

expected to deviate much further from normal with islet transport into the blood stream delayed by a much longer time, thereby potentially causing hypoglycemia. Under these circumstances, the results in Figs 5d and 5e cannot be taken to demonstrate that the behavior of thicker devices

would be satisfactory. Moreover, if the device is made even thicker than the five islet layers in Model 4, the dynamics would stretch out for even longer times, making the problem much worse. Consequently, this important issue can only be investigated with IPGTT experiments carried out with devices as thick or thicker than Model 4. The IPGTT results presented are inadequate to indicate if a large clinical device could be successful, and the analysis provided herein suggests that it could not.

6. There are two issues associated with the computational models.

A. All assume a fixed pO₂ of 40 mmHg at whatever surfaces are O₂ permeable. This may be unrealistic: (1) There will be a gradient from that surface to the nearest source of oxygen. (2) That gradient will be increased by the presence of any foreign body response (usually worse than for a mouse in larger animals) at the device-tissue surface as well as the presence of any respiring cells at that surface. The presence of a significant diffusion resistance could lead to a lower pO₂ at the surface. Sensitivity to the surface pO₂ should be examined in numerical simulations.

B. All of the simulations appear to be carried out with a cylindrical shape with the smallest configuration. This biases the results toward success for O₂ supply. To house 500,000 islets, the length would have to be at least 700 cm long, which is impractical. All other configurations would require a larger length scale for the distance of the central islets to the periphery, thereby adding additional diffusion resistance, even for oxygen. O₂ supply for the minimal configuration examined is barely enough to sustain function and not enough to maintain viability for a small but significant fraction of islet tissue (Fig. 4j). This is a problem because death of any tissue leads to degradation of proteins by liberated proteases, followed by shedding of immuno-stimulating polypeptides that

diffuse outside the device, leading to an immune response, which can be serious if the amount shed is large enough. In effect, the device modelled is barely on the edge of practicality with respect to oxygen. Therefore, to provide a convincing case of suitable O₂ supply, it is essential that experiments and/or numerical model simulations be carried with larger devices akin to clinical size containing much greater amounts of islets.

The approach taken in this study is to examine experimental or theoretical model conditions that favor success to the point where some of the results are misleading. It is essential that the manuscript explicitly discuss the limitations of any results and problems that could arise in application to much larger clinical devices so that the reader is provided a clear understanding of the limitations.

Additional (Minor) Issues

1. Lines 92-94 should be deleted or restated. They are misleading because they deal only with part of the problem in islet encapsulation (O₂ supply), but the demonstrated solution creates a problem with insulin delivery that likely renders the device unusable in diabetes.

2. The supply of oxygen through permeable outer surfaces of SONIC in contact with tissue or body spaces should be explicitly stated early in the manuscript.

Such a statement is missing now, leaving the reader to ponder where the oxygen/air comes from until one views the diagrams in the Supplement.

3. The manuscript makes it difficult to know what dimensions of SONIC are being used in models and experiments and the concentration of cells or islets in the gel regions. Information is now spread out in text, Fig. captions, Methods, and Supplement, or not at all. All of the information should be collected and systematically tabulated in one place or consistently placed where appropriate.

4. The sentence in lines 493-495 is incorrect. The permeability of perfluorocarbons is in fact much higher than in hydrogels because its O₂ solubility is much higher. The limited improvement to permeability in a PFC emulsion stems from the fact that it is an emulsion where the PFC phase is not continuous, and the modest improvement follows from the physics of permeation through heterogeneous media where the dispersed phase has the higher permeability.

Reviewer #3:

Remarks to the Author:

In this manuscript, Ma and coworkers described the design of a biomimetic scaffold featuring internal continuous air channels endowed with 10,000-fold higher oxygen diffusivity than hydrogels. The scaffold facilitates rapid oxygen transport through the whole system cells several millimeters away from the device-host boundary. Overall, it is an excellent piece of work. I support the acceptance of this work after addressing the following minor issues.

1. I am curious that whether the enhanced oxygen transportability is contributed by the structure or the fluoropolymer of the SONIC scaffold. Please further discuss this.
2. During the oxygen concentration measurement in Figure 3, the samples (only three dots) for the standard curve building is deemed insufficient. Please add at least five more dots in the standard curve, especially between the 0-40 pO₂ (mmHg).
3. It would be better to change the color of PNDf-HFP from blue to green in Figure 1k. That would be easier to observe whether the PVDF-HFP region overlaps with the area of air.
4. Is the SONIC scaffold degradable in the body? What would happen if this scaffold is implanted in the mice without retrieval?

Point-by-point reply to reviewer comments, Manuscript No. NCOMMS-20-49607-T
(All responses were colored in blue, and all changes in the manuscript were highlighted in yellow)

A bioinspired scaffold for rapid oxygenation of cell encapsulation systems

Long-Hai Wang¹, Alexander Ulrich Ernst¹, Duo An¹, Ashim Kumar Datta¹, Boris Epel²,
Mrignayani Kotecha³, and Minglin Ma^{1*}

¹Biological and Environmental Engineering, Cornell University, Ithaca, NY 14853, USA

²Department of Radiation and Cellular Oncology, The University of Chicago, Chicago, IL, 60637, USA

³O2M Technologies, LLC, Chicago, IL, 60612, USA

*Corresponding author. E-mail: mm826@cornell.edu

Reviewer #1

Remarks to the Author:

This is an impressive study performed by Wang et al, who have designed a material to facilitate greater O₂ transport to cells embedded in hydrogels. Following a bio-inspired design, the authors have designed a material to enable O₂ transport in the gas phase throughout bicontinuous, non-wettable micro-channels in PVDF-HFP scaffolds. The material does not generate oxygen. Rather, it enables oxygen transport from the ambient surroundings to cells encapsulated in hydrogel that is solidified around the scaffold. Using this strategy, the authors have demonstrated that they can provide a more spatially uniform oxygen tension throughout the hydrogel and improve cell viability and long-term in vivo efficacy.

In the study, the authors first characterized the material structure, wettability properties, and oxygen transport properties. They demonstrated that the oxygen transport properties were superior to that of a PLA scaffold control. The authors then compared the viability of cells encapsulated in alginate hydrogel around their scaffold to that of cells in alginate hydrogels only and showed that the presence of the hydrogel increased cell viability. Similarly, they showed that alginate hydrogels encapsulating islet cells and implanted in vivo without their scaffolds were not able to treat diabetic mice for extended periods, whereas those with their scaffolds could treat diabetic mice for up to 6 months (and it was reversed when the scaffold was removed). Overall, the authors present robust characterization that demonstrates their scaffold enhances oxygen transport from the surroundings and promotes the therapeutic benefit of islet cells.

One critique of the study is the controls chosen. The solution to oxygen transport is premised upon the non-wettability of the continuous microchannels (that facilitated gas phase O₂ transport) combined with the wettability of the scaffold exterior that enabled hydrogel encapsulation around the material. The controls for the study were not chosen to demonstrate that it is this particular combination of features that provides the advantage. The controls used were either solid PLA scaffold or no scaffold. It would be more convincing to demonstrate that it is not simply the microporous polymer that provides the benefit. Specifically, there is no ambient gas phase O₂ in the peritoneum, therefore the presumption is that O₂ must come out of solution to be transported in the gas phase to encapsulated cells.

Response. We thank the reviewer for his or her keen evaluation of the manuscript and favorable overall assessment. The reviewer raises an important critique about the selection of control devices, noting that the possibility of the PVDF-HFP polymer itself being responsible for the observed improvement was not explicitly ruled out. Oxygen transport in the scaffold is dependent on its oxygen permeability, which is given by the product between the oxygen solubility, α , and oxygen diffusivity, D . However, PVDF-HFP itself is a semicrystalline copolymer which has a low oxygen permeability similar to that of the PLA control (refs: Cardoso et al., *Polymers*, 2018, 10, 161; El-Hibri & Paul, *Journal of Applied Polymer Science*, 1986, 31, 2533).

We may thus evaluate the possibility of PVDF-HFP being responsible for improved oxygen transport by comparing its oxygen permeability, (αD), to that of PLA (the control used in the manuscript) and of the SONIC scaffold. We consider two alternative controls to PLA: a solid PVDF-HFP scaffold and a microporous PVDF-HFP control filled with alginate. Oxygen permeabilities for the various potential scaffold materials are listed in **Table R1**, which has been added as **Table S1 in the revised manuscript**.

Table R1: O₂ solubility, α , diffusivity, D , and permeability, (αD) in various potential scaffold materials.

Material	α	D	(αD)
	(mol/m ³ /Pa)	(m ² /s)	(mol/m/s/Pa)
PLA	4.50×10 ⁻⁵	1.60×10 ⁻¹²	7.20×10 ⁻¹⁷
Solid PVDF-HFP	3.29×10 ⁻⁵	3.50×10 ⁻¹²	1.15×10 ⁻¹⁶
Porous PVDF-HFP/alginate [†]	1.64×10 ⁻⁵	1.89×10 ⁻⁹	3.10×10 ⁻¹⁴
SONIC	3.90×10 ⁻⁴	1.80×10 ⁻⁵	7.02×10 ⁻⁹

[†]The coefficients for the porous PVDF-HFP/alginate were calculated by the composition volume fraction-weighted average of the coefficients for PVDF-HFP and alginate.

A comparison reveals that the oxygen permeabilities in PVDF-HFP and PLA are quite similar, and both several orders of magnitude lower than that expected in the SONIC scaffold, owing mostly to the significantly slower oxygen diffusivity in the solid phase. We therefore do not expect that the microporous PVDF-HFP itself is responsible for improved oxygen distribution in the SONIC system. We tested this effect of scaffold permeability on cell oxygenation and survival in the computational model (Fig. R1).

Fig. R1. Computational comparisons of scaffold type on the effect of expected oxygenation of rat islets in the SONIC device system. Expected mean population pO₂ (A) and necrotic fraction (B) of 500 IEQ of rat islets in the device featuring a scaffold comprised of PLA, solid PVDF-HFP, porous PVDF-HFP/alginate, or the SONIC scaffold. (Statistics: two-way ANOVA followed by Sidak's test for multiple comparisons. Significance: A and B: **** $p < 0.0001$).

The results of Fig. R1 show virtually no difference between oxygenation and necrosis of islets in a device containing PLA or solid PVDF-HFP, compared to a marked difference to those in the SONIC device. This is an important point to clarify to our readers. An adapted version of Fig. R1 has been added to the Supplementary Materials as Fig. S7d,e in the revised manuscript, with accompanying text in the section, “**The SONIC device improves cell survival under hypoxic conditions**”:

“Here, SONIC’s advantageous O₂ delivery mechanism is illustrated: external O₂ crosses only a thin barrier of the slow-diffusivity alginate before it reaches the SONIC scaffold, where it permeates rapidly throughout the structure, achieving a scaffold pO₂ level near

that of the surrounding environment (Fig. 1d). This rapid equilibration is achieved because of the high O₂ permeability of the SONIC system, which is enabled by the bicontinuous air channels rather than the PVDF-HFP material itself (Fig. S7d,e and Table S1).” (Page 9)

Closing Remarks

We thank the reviewer for his or her helpful critique of this manuscript. He or she identified an important unresolved question, which we hope we have addressed in full in our response. The additional analysis we performed in response to this reviewer’s critique have undoubtedly improved the rigor and quality of this work.

Reviewer #2

Remarks to the Author:

This study describes a novel approach for cell therapy of diabetes by using a porous scaffold (akin to closely connected segments of porous tubes) to bring air in close proximity in situ to islets encapsulated in a gel. The oxygen diffuses through the gas in the porous space of the scaffold. The air is supplied through external surfaces in contact with tissue or fluid in the peritoneal space. The method described is a significant variation of the notion of delivering oxygen in situ to encapsulated cells, rather than solely through the adjacent microvasculature, which was described originally more than 20 years ago and manifested more recently in designs that utilize islets encapsulated in a planar configuration exposed to elevated oxygen supplied exogenously. The main differences are the way in which the oxygen is supplied from the surrounding tissue through the porous structure instead of pumping oxygen into the device from an implanted or external source.

The paper gives convincing evidence that adequate, rapid oxygenation over the entire domain of islets is achieved in the experiments carried out. The methods are comprehensive and the results mostly convincing. However, the approach suffers from flaws that render it impractical for actual use, and the configuration used in some of the experiments is likely not representative of what is achievable in a clinical device.

Response. We thank the reviewer for his or her thorough evaluation of the manuscript and for his or her identification of areas in need of further clarification and analysis. It is evident, from the quality of the reviewer's critique, that he or she is very knowledgeable in the field of device-enabled islet delivery, especially so with respect to the literature regarding oxygen transport in such systems. We acknowledge that more work is required for such a device to be used in a clinical setting. To mitigate the reviewer's concern, we have included a new analysis (and accompanying Fig. 7 in the revised manuscript) which explores a new design to improve clinical feasibility in islet encapsulation systems. We also reemphasize that this approach has broader utility for cell delivery applications especially for those that only require constitutive (rather than phasic) therapeutic release, such as for liver disease and hemophilia. Our responses to each of the reviewer's questions are provided below.

Major Issues. In proposing a new device concept for therapeutic use, it is essential to consider the ultimate utility in for the design of a clinical device and the limitations and potential problems that might occur in achieving required function of the device, which is not done in the manuscript. In addition, the results of this study should be compared to that of other approaches in the literature that use in situ supply of oxygen.

Response. The reviewer rightly points out that our initial draft did not sufficiently acknowledge and address the limitations and potential problems for clinical applications. We have now included an analysis that explores a more clinically feasible design and added more discussions to compare this system with other approaches. We are fully aware of (and highly admire) the wonderful work that has been done in the supply or *in situ* generation of elevated oxygen levels to encapsulation devices; the work has convincingly demonstrated the importance of oxygenation to the final success of cell encapsulation despite remaining challenges such as requirements of refilling/injections (in case of the β Air approach) or battery re-charging/replacement (in the case of the electrochemical approach). On the other hand, we view the SONIC scaffold approach presented herein as a passive way to mitigate challenges related to oxygen supply, which, while it does not provide suprphysiological islet oxygenation, does not require complicated additional reactions or modules (*i.e.*, it makes the best use of the oxygen naturally available to the system). Importantly, we also believe that the strategies are not mutually exclusive: the SONIC scaffold

may be used to improve oxygen transfer in exogeneous oxygen supply systems as well. Nonetheless, we view this critique as serious, and have accordingly extended the manuscript to include an exploration of a more clinically feasible device configuration enabled by the SONIC scaffold, which also obviates the long delay times for insulin release as would be expected by simply extruding the ladder-like structure in 3 dimensions (e.g., as in the design presented in Fig. 6). As mentioned above, we also added and modified text to reemphasize that this technology is not limited to islet delivery and may be even more suitable for cell therapy applications for the treatment of liver disease and hemophilia, which do not require phasic therapeutic release. We note that it is for this reason we entitled the manuscript “**A bioinspired scaffold for rapid oxygenation of cell encapsulation systems**”, and we have adapted the text to better reflect this intention. We also better contextualize the SONIC approach among other strategies for overcoming oxygen limitations, including *in situ* O₂ supply. A corresponding description has been added to the “**Introduction**” section and is reproduced in our response to **Comment 1**.

1. The literature review is incomplete. Previous studies with configurations in which O₂ supply is enhanced and not limited to passive diffusion from surrounding blood vessel is barely mentioned. This study should be placed in the context of and compared to, other studies using in situ supply of oxygen, in particular those that supply (elevated) oxygen within the device to cells encapsulated in a planar configuration. Examples include Canadian Patent Application 2924681 and refs 101 and 107 cited in Ref 14 of this manuscript.

Response. As mentioned above, we fully agree with the reviewer that the manuscript could be improved with additional passages which contextualize our approach amongst other efforts to overcome oxygen limitations in islet delivery systems. The “**Introduction**” section was amended to include such a passage with all above references cited, which is reproduced below:

“A thoroughly investigated approach to improve graft oxygenation is to supply exogeneous O₂ in situ. The βAir device (Beta-O₂), for example, supports injections of high concentration O₂ into a gas-permeable chamber adjacent to hydrogel-encapsulated cells^{13, 14}. Another strategy is the local production of O₂ using chemical reactions^{15, 16} or electrolysis^{17, 18}. Though these strategies have all demonstrated the benefit of adequate O₂ supply to encapsulated cells, remaining limitations include increased device complexity and the requirement of patient compliance to maintain O₂ provision. O₂ transport in hydrogels is invariably dependent on its permeability, the product of the solubility and diffusivity coefficients, both of which are low in aqueous media such as hydrogels and tissue. An alternative, possibly simpler or complementary approach is thus to improve the O₂ permeability of the encapsulating material.” (Page 1,2)

Again, we note that the SONIC system is not best understood exclusively as a possible alternative to “active” oxygen supply systems, some strategies of which are also under exploration in our lab. Instead, we envision that the SONIC system even may be used to complement other “active” delivery strategies to improve oxygen transport from its site of generation/supply to the encapsulated cells, or to enhance the distribution of oxygen within the encapsulation matrix as done in this work. We convey this in the revised “**Discussion**”:

Regardless of design, the current SONIC system is reliant on the O₂ available in the transplantation site (Fig. S12); however, we also posit that the SONIC scaffold could be used to further enhance O₂ supply to hydrogel encapsulated cells even in devices which provide exogeneous supply. (Page 17)

We thank the reviewer for this helpful comment, which has encouraged us to better contextualize our approach, certainly for the benefit of the manuscript.

2. The maximum islet concentration in the gel regions examined, based on data provided for both islets and cells, is estimated to be less than 0.01 (v/v). On a macroscopic basis (including the entire volume), the islet density is exceedingly low (volume fraction of about 0.003). An islet dose of 500,000 islets requires a volume of about 0.89 ml leading to a total device volume of about 300 ml. A clinical device could be arrayed in a variety of ways by varying shape, dimensions, and total surface area, leading to large length scales (thickness), and/or very large area, but there is no conceivable effective arrangement that would not be a surgical nightmare to implant this large volume for a clinical device in one or more modules in the peritoneum or in any body space for that matter.

Response. The reviewer correctly indicates that if the specifications of the rodent model were applied to a clinical device, containing an islet payload sufficient for therapeutic effect in a human patient would not be feasible. However, we expect that even in the rodent model containing rat islets with notably high respiration rates, much higher cell densities may be applied without significant impact on cell survival. For example, Fig. S6e shows minimal expected impact of increasing dispersed cell density in the SONIC system, in contrast with a drastic effect observed in a scaffold-free control. We tested this hypothesis with rat islets in the computational model, assessing the performance of a short SONIC device (6.4 mm in length, versus 20.4 mm) containing the mouse-curative dose of 500 IEQ rat islets, marking a ~3.2-fold increase in rat islet density (Fig. R1, also added as Fig S10 in the revised manuscript).

Fig. R1. Simulation-predicted performance of a SONIC device with 500 IEQ rat islets in cylindrical devices at different device lengths (and thus cell densities). **A,B**, Annotated schematics of the 20.4 mm length SONIC device (**A**) and the 6.4 mm length SONIC device (**B**). **C**, Boundary conditions: a constant pO₂ of 40 mmHg was applied to all surfaces of both devices. **D,E**, Simulation predictions of the mean islet population pO₂ (**D**) and necrotic fraction (**E**) of islet tissue within control devices (blue) versus SONIC devices (green) of the

specified lengths. (Statistics: two-way ANOVA followed by Sidak's test for multiple comparisons. Significance: D: **** $p < 0.001$, all comparisons. E: n.s. $p > 0.05$ for SONIC device, 20.4 mm versus 6.4 mm; **** $p < 0.001$, all other comparisons).

As shown in Fig. R1 above, increasing rat islet density >3-fold should not be expected to drastically worsen functional outcomes of the devices. More importantly, literature reported values indicate that, despite batch-to-batch variation, rat islets generally have an oxygen consumption rate (OCR) that is much greater than that of human islets (0.0340 mol/m³/s versus 0.0134 mol/m³/s; refs: Avgoustiniatos *et al.*, *Ind. Eng. Chem. Res.*, 2007, 46, 6157 and Papas *et al.*, *Am. J. Transplant.*, 2007, 7, 707). Therefore, it may be hypothesized that a SONIC device could support a significantly higher loading density of human islets.

We tested a SONIC device containing a range of human islet densities up to ~8% (volume of islets per volume of device) versus an empty control with the same volume of islets in the computational model (Fig. R2, also added as Fig. S11 in the revised manuscript).

Fig. R2. Simulation-predicted performance of a SONIC device with human islets at variable loading density (volume of islets per volume of device). **A**, Annotated schematic depicting the dimensions of the SONIC device containing human islets at variable densities. **B**, Boundary conditions: a constant pO_2 of 40 mmHg was implemented on all faces. **C**, (Number basis) probability density function (f_N) of the size distribution of human islets (from Buchwald *et al.*, *Cell Transplantation*, 2009, 18, 1223). **D**, Schematic illustrating the SONIC device and empty control device encapsulating increasing densities of human islets. **E,F**, Simulation predictions of the mean islet population pO_2 (**E**) and necrotic fraction (**F**) of human islets in empty control devices (blue) versus SONIC devices (green) at the tested

densities. Densities were calculated as volume of islets per volume of device. (Statistics: two-way ANOVA followed by Sidak's test for multiple comparisons; $n = 3$ for all studies. Significance: **** $p < 0.001$; * $p < 0.05$).

The results presented in Fig. R2 show that negligible necrosis is expected for a SONIC device containing 5.44% (v/v), and relatively low necrotic levels at 6.80% and 8.02%. We note that these simulation results do not seem unreasonable as other devices which do not provide exogenous oxygen supply have demonstrated islet viability at similar densities. For example, survival and function of human islets at 7% (v/v) initial loading density and stem cell-derived β cells at 11% (v/v) were observed in 1 mm diameter cylindrical devices up to months after implantation in the peritoneal space of mice (ref. Wang *et al.*, 2021, *Sci. Transl. Med.*, 13, eabb4601).

Therefore, for application of the SONIC platform in the clinic, we expect that cell densities significantly higher than those used for rodent studies herein may be used. Specifically, at the loading density of 7%, the total volume of a device containing an islet volume of 500,000 IEQ would be around 12.6 mL, which is considerably more realistic.

We concede that the SONIC scaffold occupies significant "dead" space in its current configuration, but we have shown in Figs. S22 and S23 that the scaffold can be fabricated into many shapes other than ladder-like one inspired by the trachea studied herein. A particularly feasible arrangement is a planar design featuring interlocking spirals of hydrogel and SONIC scaffold, which would also allay concerns related to the delay in insulin release. This arrangement is explored further in our response to **Comment 6B**.

3. There is no provision for an immunobarrier of any kind. This greatly reduces the value of using an encapsulation device because immunosuppression is still required for both allogenic and xenogenic cell sources. Provision of such a necessary barrier would add additional diffusive resistance for both oxygen and insulin.

Response. We agree that the application of a nanoporous membrane between the encapsulated cells and the host site would be preferable for immune exclusion, at the cost of additional diffusive resistance to the relevant species as the reviewer indicates. However, reasonable evidence in the literature which suggests that the alginate hydrogel alone provides sufficient immunoprotection for allogeneic and even xenogeneic tissue in some animal models (ref: Bochenek et al., Nature Biomedical Engineering, 2018, 2, 810). Moreover, **Diatranz Otsuka Limited** recently completed a Phase I/II clinical trial using alginate capsules (DIABECCELL®, the initial technology licensed from **Living Cell Technologies**) for xenotransplantation without the use of immunosuppressive drugs (ClinicalTrials.gov identifier: NCT01739829; <https://clinicaltrials.gov/ct2/show/NCT01739829>). Regardless, we note that the SONIC scaffold can be fabricated into different geometries to fit into a tubular or planar chambers for additional immune exclusion or vascular stimulation (such as the ViaCyte membrane).

4. Because of the large total volume and the large fraction of volume devoted to the insulin-impermeable porous oxygen supply scaffold, the diffusion path for secreted insulin is extremely large for any configuration. The successful oxygenation of islets kept far from vessels produces a major flaw for dynamic supply of insulin to the bloodstream. The secreted insulin must diffuse sufficiently rapidly through the device in order to achieve glucose homeostasis. Hypoglycemia is a potential result of too slow response, and it can lead to coma and death. Estimates in the literature suggest a time constant for the response to a step change in BG should be on the order of 15 minutes or less in humans, which likely cannot be met with this configuration. The time constant for the scaffold device for the simplest case of Model 4 with a half thickness of 0.33 cm

and an insulin diffusivity of about 1×10^{-6} cm²/s is $t \sim L^2/D = 100,000$ s (about 30 h), which indicates that the insulin secreted from the islets at the center of the device would take an exceedingly long time to reach the bloodstream no matter what specific design is used, leading to a very dangerous situation with a device that could not fulfill its primary function.

Figures 5d and 5e show mouse IPGTT responses that are not normal but are closer to normal than to control devices (without scaffold), which at first suggests that the dynamics of insulin secretion are not terribly abnormal with the device. However, these results are misleading because they were obtained with the thin device that contains only three layers of islets. (Alternatively, the specific small device used can be viewed macroscopically as a cylindrical geometry.) When viewed as a layered device, the top and bottom layers sit immediately adjacent to the device exterior (host tissue), which means that the insulin from these islets needs to diffuse only a small distance (to nearest blood vessels). Thus, the near-normal IPGTT results from the rapid response of the insulin from 2/3 of the islets in the device (if the structure is a square.). In the thicker device with 5 layers of islet in a planar configuration, the surface layers constitute only 40% of the islet contents, so the response would be expected to deviate much further from normal with islet transport into the blood stream delayed by a much longer time, thereby potentially causing hypoglycemia. Under these circumstances, the results in Figs 5d and 5e cannot be taken to demonstrate that the behavior of thicker devices would be satisfactory. Moreover, if the device is made even thicker than the five islet layers in Model 4, the dynamics would stretch out for even longer times, making the problem much worse. Consequently, this important issue can only be investigated with IPGTT experiments carried out with devices as thick or thicker than Model 4. The IPGTT results presented are inadequate to indicate if a large clinical device could be successful, and the analysis provided herein suggests that it could not.

Response. We are grateful for and impressed by the detailed analysis provided by this reviewer. We acknowledge that the issue of delayed insulin release in devices with multi-millimeter length scales is indeed a limitation of the extended SONIC device for applications in type 1 diabetes. We have implanted 5 additional thick ($6.6 \times 6.6 \times 6.6$ mm) SONIC devices in mice, each containing 500 IEQ of rat islets, and performed an intraperitoneal glucose tolerance test (IPGTT) with the thick devices as suggested by the reviewer, indeed finding a delay in blood glucose correction compared with healthy mice (Fig. R3, also added as Fig. 6i,j in the revised manuscript).

Fig. R3. Results from additional *in vivo* studies with the 6.6 mm SONIC devices. **A**, Blood glucose (BG) measurements of diabetic C57BL6/J mice following IP transplantation of the 6.6 mm SONIC devices (pink, $n = 5$, retrieved on day 60; red, $n = 5$, retrieved on day 120). **B**, IPGTT on day 58; mean \pm SD. (Statistics: two-way ANOVA followed by Sidak's test for multiple comparisons; Significance: $**p < 0.01$, 6.6 mm SONIC device-treated mice versus healthy mice).

We have articulated this limitation to the readers in the section, “**The SONIC system enables diabetes correction in mice with thick device**”:

“Rat islets (500 IEQ per transplant) were incorporated in devices and transplanted into the intraperitoneal cavity of diabetic C57BL/6 mice (n = 10), with 5 retrieved at day 60 and the remainder retrieved at day 120 (Fig. 6g,h). Strikingly, the mice achieved normoglycemia within a few days after transplantation, with 4 out of the 5 mice in the long-term study maintaining normoglycemia over 4 months (Fig. 6i). Additionally, the BG returned to hyperglycemia after device retrieval, confirming the role of the device in diabetes correction. An IPGTT performed on day 58 showed that the devices restored normoglycemia within 120 min, though the profile in device-treated mice showed a statistically significant delay in comparison to healthy control mice (Fig. 6j). BG monitoring was extended for an additional 1 h to monitor potential hypoglycemia; 6.6 mm SONIC treated mice showed a slight overcorrection to ~70 mg/dL at 150 min (compared to ~120 mg/dL in healthy control mice) but stabilized near this value at the 180 min time point. The observed response delay and hypoglycemic overcorrection is likely because of the significant diffusion distance for insulin secreted from deeply encapsulated islets.” (Page 14)

To address or mitigate the reviewer’s concerns in this regard, we wish to respond in two different ways:

- (1) We hope to reemphasize the broader utility of the SONIC concept for other types of cell replacement therapies. As mentioned in the manuscript introduction, the SONIC system may also be used for the delivery of cells for the treatment of liver diseases and hemophilia. Both applications also require the delivery of cell clusters in high payloads, and unlike diabetes, they do not require dynamic factor release, thus the diffusion distance for therapeutic release is less of a concern. For example, in a Phase I clinical trial for Hemophilia A treatment, a subclinical dose of 400 million non-encapsulated factor VIII-secreting cells were transplanted into the omentum (ref: Roth *et al.*, *N. Engl. J. Med.*, 2001, 344, 1735). Recently, **Sigilon Therapeutics** also reported the transfer of their islet encapsulation system (modified alginate capsules) for the treatment of Hemophilia A by encapsulating factor VIII-secreting cells, initiating a first-in-human Phase I/II clinical trial with this technology (ClinicalTrials.gov identifier: NCT04541628; <https://clinicaltrials.gov/ct2/show/NCT04541628>). In this study, **Sigilon** intends to implant 100-150 mL of alginate microcapsules containing 2×10^9 to 3×10^9 cells into the intraperitoneal cavity of patients, highly resembling values for islet delivery (<https://sec.report/Document/0001558370-21-003212/>). This notion is conveyed in a sentence in the “**Introduction**” and in a new paragraph in the “**Discussion**”:

Clinical islet transplantations require approximately 500 k islet equivalent (IEQ) of human islets (5 k–10 k IEQ per kg body weight) to reverse diabetes¹, and cellular treatments for liver diseases and hemophilia require similar cell volumes⁵. (Page 2)

“A successful islet delivery implant must not only maintain cell survival but also ensure timely release of insulin to prevent postprandial hyperglycemia and overcorrection into hypoglycemia. The delay in IPGTT observed in mice treated with the thick SONIC device (Fig. 6j) indicate that, for islet delivery, simply extending the SONIC device’s structure in all three dimensions may not be tenable. However, cellular treatments for hemophilia and liver diseases require similar cell payloads⁵ without requiring phasic therapeutic release. Therefore, the thick SONIC system may be used to overcome capacity limitations in these applications.” (Page 18)

- (2) Furthermore, we believe that the feasibility of a clinical scale device enabled by the SONIC scaffold may be improved by a new configuration, which may obviate the concern of insulin diffusion delay. This configuration is discussed in detail in our response to **Comment 6B**.

Again, we are grateful to the reviewer for providing this critique as it is critical that this is addressed in the manuscript.

6. There are two issues associated with the computational models.

A. All assume a fixed pO₂ of 40 mmHg at whatever surfaces are O₂ permeable. This may be unrealistic: (1) There will be a gradient from that surface to the nearest source of oxygen. (2) That gradient will be increased by the presence of any foreign body response (usually worse than for a mouse in larger animals) at the device-tissue surface as well as the presence of any respiring cells at that surface. The presence of a significant diffusion resistance could lead to a lower pO₂ at the surface. Sensitivity to the surface pO₂ should be examined in numerical simulations.

Response. The reviewer correctly indicates that the pO₂ on the device surface (P_{ext}) may not be exactly 40 mmHg, as applied in the models of the manuscript. Literature reported pO₂ measurements in common transplantation sites have shown significant variance. In nonhuman primates, intraperitoneal pO₂ measurements ranged from 20–50 mmHg in one study (ref: Bochenek et al., Nature Biomedical Engineering, 2018, 2, 810), but another measurement pins this range to 61 ± 11 mmHg using noninvasive fluorine-19 magnetic resonance relaxometry and to 89 ± 6.1 mmHg using a fiber-optic oxygen sensor (Safley et al., Transplantation, 2020, 104, 259). In mice, intraperitoneal pO₂ has been measured at 44.9 ± 6.8 mmHg (Bourdel et al., Human Reproduction, 2007, 22, 1149) and has also been reported at 51–58 mmHg (Papapoulos et al., Adv Drug Deliv Rev, 2019, 139, 139). Furthermore, many modeling studies have used $P_{ext} = 40$ mmHg as convention (refs: Colton et al., Chemical Engineering Science, 2009, 64, 4470; Dulong and Legallais, Biotechnol. Bioeng., 2007, 96, 990; Buchwald et al., Biotechnol. Bioeng., 2018, 115, 232; Buchwald et al., Biotechnol. Bioeng., 2015, 14, 28). For these reasons, we felt that 40 mmHg was an appropriate and moderate value for P_{ext} which would not overstate the external oxygen level of the system.

As the reviewer indicates, a pO₂ gradient in nearby tissue or the presence of a fibrotic capsule may result in a surface oxygen level, P_{ext} , to be lower than 40 mmHg, but the cited measurements also suggest that P_{ext} may also be higher than 40 mmHg. Nevertheless, we followed the reviewer's suggestion and determined the sensitivity of islet oxygenation and cell survival to P_{ext} at values of 24, 32, 40, and 60 mmHg (**Fig. R4**, also added as a part of **Fig S12 in the revised manuscript**).

A Boundary conditions:

Fig. R4. Effect of variable external pO₂ on simulated rat islet (500 IEQ) oxygenation in the SONIC device (4.2 mm diameter, 20.4 mm in length) versus the control device. **A**, Boundary conditions: a series of pO₂ values were implemented on all faces. **B,C**, Simulation predictions of the mean islet population pO₂ (**B**) and necrotic fraction (**C**). (Statistics: two-way ANOVA followed by Sidak's test for multiple comparisons. Significance: **** $p < 0.0001$; * $p < 0.05$).

Though we naturally see that lower P_{ext} leads to lower oxygenation and survival in both the control and SONIC devices, the presence of the SONIC scaffold significantly mitigates these poor outcomes. In any case, as the SONIC system does not feature exogeneous supply, it is exclusively dependent on the $p\text{O}_2$ available in the local environment. We have made sure to express this to the readers in the “**Results**” and the “**Discussion**”:

...and mitigate negative outcomes should the external $p\text{O}_2$ environment be lower (Fig. S12) (Page 10)

Regardless of design, the current SONIC system is reliant on the O_2 available in the transplantation site (Fig. S12); however, we also posit that the SONIC scaffold could be used to further enhance O_2 supply to hydrogel encapsulated cells even in devices which provide exogeneous supply. (Page 17)

We are grateful for the reviewer for encouraging us to pursue this analysis.

B. All of the simulations appear to be carried out with a cylindrical shape with the smallest configuration. This biases the results toward success for O_2 supply. To house 500,000 islets, the length would have to be at least 700 cm long, which is impractical. All other configurations would require a larger length scale for the distance of the central islets to the periphery, thereby adding additional diffusion resistance, even for oxygen. O_2 supply for the minimal configuration examined is barely enough to sustain function and not enough to maintain viability for a small but significant fraction of islet tissue (Fig. 4j). This is a problem because death of any tissue leads to degradation of proteins by liberated proteases, followed by shedding of immuno-stimulating polypeptides that diffuse outside the device, leading to an immune response, which can be serious if the amount shed is large enough. In effect, the device modelled is barely on the edge of practicality with respect to oxygen. Therefore, to provide a convincing case of suitable O_2 supply, it is essential that experiments and/or numerical model simulations be carried with larger devices akin to clinical size containing much greater amounts of islets.

The approach taken in this study is to examine experimental or theoretical model conditions that favor success to the point where some of the results are misleading. It is essential that the manuscript explicitly discuss the limitations of any results and problems that could arise in application to much larger clinical devices so that the reader is provided a clear understanding of the limitations.

Response. We thank the reviewer for raising this important critique. The scalability and therefore feasibility of the SONIC device as a potentially clinical cell delivery system are not sufficiently discussed. Acknowledging that an alternative design may be necessary for clinical feasibility, we performed additional simulations as suggested by the reviewer. Unfortunately, computational limitations of our modeling workstation preclude us from carrying out simulations with hundreds of thousands of islets. However, we may simulate a section of a device and extrapolate the results to infer the feasibility of a larger structure. In summary, the results suggest that the SONIC system may be modified and optimized to produce a device with more scalability and clinical feasibility for islet delivery than previously presented. To comprehensively respond to this critique, we performed a new exploration for a consideration of a potential alternative design which naturally affords more scalability (Fig. 7 and Fig. S24 in the revised manuscript).

In the first submission, we briefly mentioned in the “**Discussion**” that 3D printing enables the SONIC scaffold to be printed in a variety of configurations, with some examples provided in the Supplementary Information (Fig. S22). We consider a planar device, which incorporates the SONIC scaffold in an Archimedean spiral configuration, as a conveniently scalable design (Fig. R5; a modified version has also been added as Fig. S24 in the revised manuscript).

Fig. R5. The scalable spiral SONIC device design. **A**, Annotated schematic depicting the dimensions of a planar device featuring a SONIC scaffold in an Archimedean spiral configuration used for simulations. **B**, Oxygen boundary conditions: a constant external oxygen tension of 40 mmHg was applied to the top and bottom faces, and a no-flux condition imposed on the lateral face to mimic negligible edge effects of a device of similar configuration but radially extended. **C**, Image of the nonuniform mesh implemented in the simulation.

This planar construct features a half-thickness of 600 μm , approximately that of the βAir device (refs: Barkai et al., Cell Transplantation, 2013, 22, 1463; Evron et al., Scientific Reports, 2018, 8, 6508), thereby hopefully allaying the reviewer's concern of delay in insulin release. This design is also advantageous because the distance between any islet and the relatively high oxygen level in the SONIC scaffold can be maintained within 250 μm . The construct shown here may be viewed as the central portion of a larger device extended radially. We tested the sensitivity of this design to a range of human islet loading densities in comparison to a control device without the spiral SONIC scaffold (**Fig. R6**, also added into Fig. 7 in the revised manuscript).

Fig. R6. Computational exploration of a SONIC spiral device for delivering a clinically relevant islet dose. **A**, Annotated schematic of the central section of a hypothetical SONIC spiral device, including the SONIC scaffold (blue) arranged in an Archimedean spiral (with the distance between turns fixed at 500 μ m) and hydrogel-encapsulated human islets (red). A thickness of 1.2 mm ensures a maximum distance of insulin diffusion of 600 μ m; a diameter of 4.75 mm was used for simulations, representing the central section of a device scaled radially to achieve a sufficient encapsulated islet payload. **B**, Schematics showing the SONIC spiral device and scaffold-free control device encapsulating 4%, 6%, 8%, and 10% human islets (volume of islets per volume of device), as tested in the simulations. **C,D**, Simulation predictions of the mean islet population pO_2 (**C**) and fraction of necrosis (**D**) of human islets encapsulated at variable densities in the SONIC spiral device and the scaffold-free control device; mean \pm SD; C: **** $p < 0.0001$ (control device versus SONIC spiral device at all islet densities); D: *** $p < 0.001$ (control device versus SONIC spiral device at 4% islet density), ** $p < 0.01$ (control device versus SONIC spiral device at 6%), **** $p < 0.0001$ (control device versus SONIC spiral device at 8% and 10% islet densities). **E,F**, Surface plots of pO_2 gradients (right) at selected cross sections (left; labelled A-A, B-B, and C-C) in the control device (**E**) and the SONIC spiral device (**F**) at 8% human islet loading density showing significantly higher pO_2 and negligible necrosis in the SONIC spiral device compared to the control device (white regions in the islets represent necrosis, and yellow arrows in C-C section indicate necrosis even in some small islets).

The computational model predicts that, at all loading densities tested, the SONIC spiral device would substantially out-perform an empty control in terms of both estimated mean islet population pO₂ levels and necrotic fractions. Importantly, negligible necrotic fractions are predicted for human islets in the SONIC spiral device at 8% loading density, whereas a necrotic fraction of 4.8±0.66% is predicted at the same density in the control. At these specifications, the total device volume required to deliver a therapeutic islet payload (500,000 IEQ) is ~11 mL, corresponding to a device diameter of 10.8 cm, which is manageable surgically both in volume and in characteristic length. We added a passage to accompany this analysis in the “**Discussion**”, reproduced below:

We also emphasize that 3D printing enables the SONIC scaffold to be fabricated in scaled-up dimensions (e.g., in multiple layers or extended in length to tens of centimeters) in a wide range of designs (e.g., toroidal, spiral) (Fig. S22, Fig. S23). An advantageous configuration of an islet delivery device is a planar hydrogel disk (1.2 mm thickness) incorporating an internal SONIC scaffold configured in an Archimedean spiral with a 500 μm gap between turns (Fig. 7 and Fig. S24a). The small thickness of 1.2 mm obviates limitations associated with delays in insulin release, and the spiral structure ensures that each islet is within 250 μm of a high pO₂ source in the SONIC scaffold (Fig. 7a). We assessed the capacity of this construct to accommodate variable densities (4–10%) of human islets in comparison to a control construct without the SONIC scaffold using the computational model (Fig. 7b). Model predictions indicate significantly higher oxygenation of islets in the SONIC spiral device relative to the control construct, and negligible necrosis levels up to 8% (v/v) human islet loading density (Fig. 7c–f). If extended radially, this construct could support a curative islet dose of 500 k IEQ within a disk approximately 11 cm in diameter. (Page 18)

Furthermore, more emphasis on the broader utilization of an extruded ladder-like device for cell delivery for alternative hormone deficiency disorders is included in the “**Discussion**” as well:

“A successful islet delivery implant must not only maintain cell survival but also ensure timely release of insulin to prevent postprandial hyperglycemia and overcorrection into hypoglycemia. The delay in IPGTT observed in mice treated with the thick SONIC device (Fig. 6j) indicate that, for islet delivery, simply extending the SONIC device’s structure in all three dimensions may not be tenable. However, cellular treatments for hemophilia and liver diseases require similar cell payloads without requiring phasic therapeutic release. Therefore, the thick SONIC system may be used to overcome capacity limitations in these applications.” (Page 18)

Again, we thank the reviewer for raising this critique, and hope that our changes and further analysis have addressed his or her concerns in full.

Additional (Minor) Issues

1. Lines 92-94 should be deleted or restated. They are misleading because they deal only with part of the problem in islet encapsulation (O₂ supply), but the demonstrated solution creates a problem with insulin delivery that likely renders the device unusable in diabetes.

Response. We thank the reviewer for reiterating this point. First, we reworded many sentences in the “**Introduction**” to clearly indicate that the thickness limitation we sought to overcome was solely with respect to oxygen supply. With respect to the sentence in lines 92–94, we simply truncated the sentence to now read as follows:

The biomimetic SONIC cell delivery system solves the problem of slow and non-penetrating O₂ transport in thick bulk hydrogels. (Pages 3)

2. The supply of oxygen through permeable outer surfaces of SONIC in contact with tissue or body spaces should be explicitly stated early in the manuscript. Such a statement is missing now, leaving the reader to ponder where the oxygen/air comes from until one views the diagrams in the Supplement.

Response. We thank the reviewer for this helpful suggestion. To address this, we updated Fig. 1d in the revised manuscript, adding text to indicate the source of oxygen, and modified the caption to communicate this more clearly. It is reproduced below (**Fig. R7**):

Fig. R7. A schematic illustrating O₂ delivery from the transplantation site into inside the cell encapsulation system through a tracheal ladder network-like SONIC scaffold.

3. The manuscript makes it difficult to know what dimensions of SONIC are being used in models and experiments and the concentration of cells or islets in the gel regions. Information is now spread out in text, Fig. captions, Methods, and Supplement, or not at all. All of the information should be collected and systematically tabulated in one place or consistently placed where appropriate.

Response. We appreciate the reviewer for addressing this issue. We have now added an independent paragraph in the Methods, under the section of "Fabrication of the SONIC device", which provides all the details of the device dimensions. Additionally, the Methods section, "Computational Modeling" was heavily adapted to state the dimensions and conditions of each simulation more clearly, with explicit references to the figures which present results associated with the discussed model.

4. The sentence in lines 493-495 is incorrect. The permeability of perfluorocarbons is in fact much higher than in hydrogels because its O₂ solubility is much higher. The limited improvement to permeability in a PFC emulsion stems from the fact that it is an emulsion where the PFC phase is not continuous, and the modest improvement follows from the physics of permeation through heterogenous media where the dispersed phase has the higher permeability.

Response. We thank the reviewer for identifying this point. The permeability in PFC is indeed much higher than in alginate (~10×), but we want to clarify that our intention was to convey that it was not as much higher as in the SONIC scaffold (~100,000×). We nonetheless agree with the reviewer that this statement is misleading and has been corrected to read as follows:

Some reports have explored the effect of perfluorocarbon (PFC) emulsion incorporation in hydrogels due to the high O₂ solubility^{56,57}, and slightly higher diffusivity⁵⁸ in PFC compared to hydrogels. However, these systems generally only yield modest benefits because of the limited improvement in O₂ permeability in composite systems where the phase with the higher permeability (in this case, PFC) is dispersed⁵⁹. The bicontinuous gas phase, endowed with extremely high permeability, incorporated into the hydrogel by the SONIC scaffold facilitates the rapid permeation of O₂ throughout the device, thereby enormously improving the effective O₂ permeability of the system. (Page 17)

Closing Remarks

We are extremely grateful to the reviewer for his or her time and providing these insightful critiques and suggestions to improve the manuscript. His or her comments motivated exploration of a new design, enabled by the SONIC technology, to possibly overcome the main limitations of SONIC as a platform device for potential clinical applications. We hope that our point-by-point responses have addressed all the reviewer's concerns. The revisions and updates to the manuscript, prompted by the reviewer, have undoubtedly improved the rigor and clarity of this work.

Reviewer #3

Remarks to the Author:

In this manuscript, Ma and coworkers described the design of a biomimetic scaffold featuring internal continuous air channels endowed with 10,000-fold higher oxygen diffusivity than hydrogels. The scaffold facilitates rapid oxygen transport through the whole system cells several millimeters away from the device-host boundary. Overall, it is an excellent piece of work. I support the acceptance of this work after addressing the following minor issues.

Response. We thank the reviewer for his or her observant evaluation of the manuscript and favorable overall assessment. The reviewer raised several important questions, some of which were also raised by other reviewers. All of the reviewer's comments are addressed in our point-by-point response below and in updates to the text and figures in the manuscript.

1. I am curious that whether the enhanced oxygen transportability is contributed by the structure or the fluoropolymer of the SONIC scaffold. Please further discuss this.

Response. We thank the reviewer for his or her keen evaluation of the manuscript and favorable overall assessment. The reviewer raises an important critique about the selection of control devices, noting that the possibility of the PVDF-HFP polymer itself being responsible for the observed improvement was not explicitly ruled out. Oxygen transport in the scaffold is dependent on its oxygen permeability, which is given by the product between the oxygen solubility, α , and oxygen diffusivity, D . However, PVDF-HFP itself is a semicrystalline copolymer which has a low oxygen permeability similar to that of the PLA control (refs: Cardoso *et al.*, *Polymers*, 2018, 10, 161; El-Hibri & Paul, *Journal of Applied Polymer Science*, 1986, 31, 2533).

We may thus evaluate the possibility of PVDF-HFP being responsible for improved oxygen transport by comparing its oxygen permeability, (αD), to that of PLA (the control used in the manuscript) and of the SONIC scaffold. We consider two alternative controls to PLA: a solid PVDF-HFP scaffold and a microporous PVDF-HFP control filled with alginate. Oxygen permeabilities for the various potential scaffold materials are listed in **Table R1**, which has been added as **Table S1 in the revised manuscript**.

Table R1: O₂ solubility, α , diffusivity, D , and permeability, (αD) in various potential scaffold materials.

Material	α (mol/m³/Pa)	D (m²/s)	(αD) (mol/m/s/Pa)
PLA	4.50×10 ⁻⁵	1.60×10 ⁻¹²	7.20×10 ⁻¹⁷
Solid PVDF-HFP	3.29×10 ⁻⁵	3.50×10 ⁻¹²	1.15×10 ⁻¹⁶
Porous PVDF-HFP/alginate[†]	1.64×10 ⁻⁵	1.89×10 ⁻⁹	3.10×10 ⁻¹⁴
SONIC	3.90×10 ⁻⁴	1.80×10 ⁻⁵	7.02×10 ⁻⁹

[†]The coefficients for the porous PVDF-HFP/alginate were calculated by the composition volume fraction-weighted average of the coefficients for PVDF-HFP and alginate.

A comparison reveals that the oxygen permeabilities in PVDF-HFP and PLA are quite similar, and both several orders of magnitude lower than that expected in the SONIC scaffold, owing mostly to the significantly slower oxygen diffusivity in the solid phase. We therefore do not expect that the microporous PVDF-HFP itself is responsible for improved oxygen distribution in the SONIC system. We tested this effect of scaffold permeability on cell oxygenation and survival in the computational model (**Fig. R1**).

Fig. R1. Computational comparisons of scaffold type on the effect of expected oxygenation of rat islets in the SONIC device system. Expected mean population pO₂ (**A**) and necrotic fraction (**B**) of 500 IEQ of rat islets in the device featuring a scaffold comprised of PLA, solid PVDF-HFP, porous PVDF-HFP/alginate, or the SONIC scaffold. (Statistics: two-way ANOVA followed by Sidak’s test for multiple comparisons. Significance: A and B: **** $p < 0.0001$).

The results of Fig. R1 show virtually no difference between oxygenation and necrosis of islets in a device containing PLA or solid PVDF-HFP, compared to a marked difference to those in the SONIC device. This is an important point to clarify to our readers. An adapted version of Fig. R1 has been added to the Supplementary Materials as **Fig. S7d,e in the revised manuscript**, with accompanying text in the section, “**The SONIC device improves cell survival under hypoxic conditions**”:

“Here, SONIC’s advantageous O₂ delivery mechanism is illustrated: external O₂ crosses only a thin barrier of the slow-diffusivity alginate before it reaches the SONIC scaffold, where it permeates rapidly throughout the structure, achieving a scaffold pO₂ level near that of the surrounding environment (Fig. 1d). This rapid equilibration is achieved because of the high O₂ permeability of the SONIC system, which is enabled by the bicontinuous air channels rather than the PVDF-HFP material itself (Fig. S7d,e and Table S1).” (**Page 9**)

2. During the oxygen concentration measurement in Figure 3, the samples (only three dots) for the standard curve building is deemed insufficient. Please add at least five more dots in the standard curve, especially between the 0-40 pO₂ (mmHg).

Response. We thank the reader for identifying this issue. We have updated the standard curve for the EPR measurement data in Fig. 3c in the revised manuscript. It is reproduced below (**Fig. R2**):

Fig. R2. Calibration curve of the OX063-d24 relaxation rate versus pO₂.

3. It would be better to change the color of PNDf-HFP from blue to green in Figure 1k. That would be easier to observe whether the PVDF-HFP region overlaps with the area of air.

Response. We appreciate this suggestion. However, according to Nature Communication's author guidelines ("[Brief guide for submission to Nature Communications](https://www.nature.com/documents/ncomms-submission-guide.pdf)"; <https://www.nature.com/documents/ncomms-submission-guide.pdf>), green and red color pairings should be avoided, so we feel that it is best to keep the color as is.

4. Is the SONIC scaffold degradable in the body? What would happen if this scaffold is implanted in the mice without retrieval?

Response. We thank the reviewer for raising this question. The SONIC scaffold is not biodegradable *in vivo*; no defects were observed in the scaffold in retrieved devices after 6-month implantation. The SONIC scaffold is comprised of PVDF-HFP, which is resistant to hydrolytic, oxidative, or enzymatic breakdown, and has even been used as a stent coating in clinical applications (Grainger, *Biomaterials Science (Fourth Edition)*, "Ch. 1.3.2C – Fluorinated Biomaterials", 2020, 125). We therefore assume that the SONIC scaffold is suitable for long-term implantation *in vivo*. We have added a sentence to the section, "**Design and fabrication of the SONIC device**", communicating this to the reader:

"Furthermore, PVDF-HFP is resistant to hydrolytic, oxidative, and enzymatic breakdown, and is therefore advantageous for use *in vivo*." (**Page 4**).

In addition, the scaffold was completely covered by alginate hydrogel, and at no points is the scaffold in direct contact with the host tissue.

Closing Remarks

We are grateful to this reviewer for his or her favorable assessment of the manuscript and for raising several points which we have addressed in full in our point-by-point responses. The updates, modifications, and additional analysis inspired by this review have undoubtedly improved the clarity and rigor of this work.

Reviewers' Comments:

Reviewer #1:

Remarks to the Author:

The authors have attempted to address the major critique of the article provided, which was that the appropriate controls were not used to demonstrate that it was not simply the microporous structure that facilitated the positive outcome. Essentially, the question remains whether any microporous polymer can be used to obtain similar results or whether the specific physicochemical properties of the device (non-wettability of the continuous channels versus wettability of the scaffold exterior) were the critical (and novel) elements. It appears that the authors misinterpreted the critique and, as such, have not responded to this critique/question.

The authors instead answered whether a solid, non-porous PVDF-HFP would have similar results. They used known or calculated values of oxygen permeability to show that the SONIC device had orders of magnitude higher permeability compared to solid PVDF-HFP. They went further to model whether cell survival in low oxygen conditions would be improved using the higher or lower permeability materials. Predictably, they calculated that cell survival would be higher with the higher permeability and have included this as a supplemental result. (This result should probably be removed given its obvious nature). To rephrase the critique, other microporous polymers exist and would also have orders of magnitude higher permeability. Would they function similarly to the SONIC?

It is hoped the authors would address the question of appropriate controls - and uniqueness of their device - more directly.

Reviewer #2:

Remarks to the Author:

The authors have done a remarkably good job in responding to every comment, and the new additions and changes to the manuscript greatly increase its credibility and potential impact. It is now ready for publication.

Reviewer #3:

Remarks to the Author:

The authors have addressed my comments.

Responses to reviewer's comments, Manuscript No. NCOMMS-20-49607A

(All responses were colored in blue, and all changes in the manuscript were highlighted in yellow)

Reviewer #1 (Remarks to the Author):

The authors have attempted to address the major critique of the article provided, which was that the appropriate controls were not used to demonstrate that it was not simply the microporous structure that facilitated the positive outcome. Essentially, the question remains whether any microporous polymer can be used to obtain similar results or whether the specific physicochemical properties of the device (non-wettability of the continuous channels versus wettability of the scaffold exterior) were the critical (and novel) elements. It appears that the authors misinterpreted the critique and, as such, have not responded to this critique/question.

The authors instead answered whether a solid, non-porous PVDF-HFP would have similar results. They used known or calculated values of oxygen permeability to show that the SONIC device had orders of magnitude higher permeability compared to solid PVDF-HFP. They went further to model whether cell survival in low oxygen conditions would be improved using the higher or lower permeability materials. Predictably, they calculated that cell survival would be higher with the higher permeability and have included this as a supplemental result. (This result should probably be removed given its obvious nature). To rephrase the critique, other microporous polymers exist and would also have orders of magnitude higher permeability. Would they function similarly to the SONIC?

It is hoped the authors would address the question of appropriate controls - and uniqueness of their device - more directly.

Response. We appreciate the reviewer for clarifying his or her critique and for continuing to encourage the resolution of this important question. We furthermore apologize for our unclear answer provided in the previous revision which did not satisfactorily address the reviewer's concerns. Essentially, it has not been ruled out that any microporous polymer may be equally suitable as PVDF-HFP for the basis as SONIC, even if the substitute material did not have the properties of (1) surface wettability and (2) bicontinuous channel non-wettability.

With regards to the wettability of the scaffold exterior, we note that this property was required for the facile application of the cell encapsulation hydrogel and is not suggested to influence the permeability of oxygen in the scaffold. The hydrophilic polydopamine coating was applied to the scaffold surface to provide a compatible interface between the hydrophobic SONIC scaffold and the hydrophilic hydrogel, allowing the latter to fully penetrate the macro-spaces inside the hierarchical scaffold. We will thus focus the remainder of the response on the importance of the second property—non-wettability of the internal bicontinuous channels—on enabling beneficial oxygen transfer in the SONIC system.

In our previous response, we used computational modeling to explore the impact on oxygen distribution in the event of the PVDF-HFP microporous polymer becoming filled with liquid, which we assumed would be the case if the internal channels were hydrophilic. (This was perhaps unclearly labelled as “Porous PVDF-HFP/alginate” in the figure legend, which we have addressed by adding more clarifying descriptions in the table and figure captions). Modeling results logically indicated that, should the air channels inside the PVDF-HFP scaffold fill with liquid, the cells encapsulated in the device would have lower oxygenation and function relative to SONIC (where the air channels are maintained). To verify the vital role of the air channels in facilitating high

oxygen permeability, we have designed two additional controls and investigated their oxygen transport efficiency.

One control was a commercial porous sponge comprised of hydrophilic melamine, which was selected as a simple example to confirm that a liquid-filled porous material would not provide benefit to oxygen transport. For the second control, we performed a two-step modification process to a PVDF-HFP scaffold to render its interior microporous channels hydrophilic, to test whether the current SONIC scaffold would also become occluded with liquid and therefore lose its rapid oxygen permeability if it was completely hydrophilic. First, ethanol was introduced into the tris buffer during incubation with dopamine to allow buffer solution penetration into the micropores of PVDF-HFP, which enabled the polydopamine to coat the microporous channels and rendered them hydrophilic. Second, the scaffold was treated with radio frequency plasma to further ensure its hydrophilicity. This treated control, which we will refer to as “hydrophilic porous PVDF-HFP” is an ideal control because it is structurally identical to the SONIC scaffold with the singular exception that the internal pore surfaces were hydrophilic.

Oxygen transport tests through these two control scaffolds were conducted using electron paramagnetic resonance (EPR) oxygen imaging by the procedure followed in Fig. 3 of the main text. Both controls were observed to be significantly slower than SONIC at equilibrating the system to exposed oxygen levels (**Fig. R1a**). Spatial pO_2 profiles further confirmed unelevated oxygen permeability in the hydrophilic porous PVDF-HFP, showing a top-to-bottom pO_2 gradient throughout the system until a steady state was achieved (**Fig. R1b**), instead of the radial gradient emanating from the SONIC insert, the latter of which enabled rapid and deep penetration to the bottom (see Fig. 3f,g). The results from these two additional controls confirm the importance of the non-wettability of the internal bicontinuous pores for providing benefit to rapid oxygen transport in the SONIC system.

Fig. R1. (a) Average pO_2 in gelatin in the container over time with the SONIC scaffold, hydrophilic porous PVDF-HFP scaffold, hydrophilic porous melamine scaffold, PLA scaffold, and empty gelatin (no scaffold) after exposure to a gas mixture with a pO_2 of 40 mmHg. (b) pO_2 distributions on a tangential plane of control sample with the hydrophilic porous PVDF-HFP scaffold at different time points (indicated by the arrows in (a)), showing a slow equilibration via a top-to-bottom gradient.

The reviewer essentially questions whether any microporous polymer may provide equal benefit to SONIC. As we have discussed above, if the porous structures cannot maintain internal non-wettable air channels, they will not provide rapid oxygen permeability. The experiments performed with the new controls in this response demonstrate that the hydrophobicity of the surfaces of the internal pores is a necessary property for preventing occlusion by liquid and by extension for enabling high oxygen permeability. Thus, with exclusive respect to facilitating rapid oxygen transfer, some other microporous material with bicontinuous channels which maintain a gas phase could provide a similar benefit to SONIC.

Nonetheless, PVDF-HFP has several intrinsic attributes which make it especially desirable for application in encapsulated cell delivery. Foremost, PVDF-HFP is an FDA-approved material used in clinically available medical devices (Grainger, *Biomaterials Science* (Fourth Edition), “Ch. 1.3.2C—Fluorinated Biomaterials”, 2020, 125). Furthermore, bicontinuous air channels are easily formed within the PVDF-HFP scaffold through the immersion precipitation process. In addition, it is compatible for use in sacrificial 3D printing, which enables significant geometry flexibility unavailable to some other materials. Finally, PVDF-HFP is biocompatible, nondegradable *in vivo*, and endowed with reasonably strong mechanical properties, which makes it suitable for long-term implantation *in vivo*.

To communicate the findings of the EPR measurements with the new and more suitable controls, and to better articulate the uniqueness of the SONIC approach, we have included the EPR experiment data in Fig. S6 with the results and their importance explained in the main text, “**Results: Rapid O₂ transport through the SONIC scaffold**” section and the “**Discussion**”:

The EPR measurement procedure was repeated with two additional control scaffolds chosen to verify the vital role of the air channels in facilitating high oxygen permeability. One control was a commercial porous sponge comprised of hydrophilic melamine, which was selected as a simple example to confirm that a liquid-filled porous material would not provide benefit to O₂ transport. The second control was a PVDF-HFP scaffold, modified by the following two step process to render the internal microporous channels hydrophilic: (1) ethanol was added to the tris buffer during incubation with dopamine to allow buffer penetration into the micropores of the scaffold, therefore enabling the application the hydrophilic polydopamine coating within the microchannels; (2) the scaffold was treated with radio frequency plasma to further ensure its hydrophilicity. O₂ transport tests showed that these two scaffolds were significantly slower than SONIC at equilibrating the system to exposed O₂ levels (Fig. S6a). Spatial O₂ distributions of the sample containing the hydrophilic porous PVDF-HFP showed a top-to-bottom pO₂ gradient throughout the system until a steady state was achieved (Fig. S6b), rather than the radial gradient emanating from the SONIC insert (Fig. 3g), the latter of which enabled rapid and deep O₂ penetration to the bottom. The results from these two additional controls indicate the necessity of the non-wettability of the internal microchannels for enabling rapid O₂ transport in the SONIC system. (Page 8)

...while maintaining internal hydrophobicity to avoid water penetration into the air channels which were verified as essential for enabling the high O₂ permeability of the scaffold. (Page 18)

We would like to thank the reviewer for persisting with his or her critique of our control selection, and for his or her patience with our misunderstanding in the first response. We hope that the new experiments performed herein have allayed his or her concerns. Again, we are grateful as additional comparisons and added text motivated by this reviewer’s comments have certainly improved the rigor of the manuscript and have aided us to better articulate the significance of this project to the paper’s audience.

Reviewers' Comments:

Reviewer #1:

Remarks to the Author:

The authors have done a great job at responding to the query and have addressed my concerns.